# AN EXACT SOLVER FOR SATISFIABILITY MODULO COUNTING WITH PROBABILISTIC CIRCUITS

## ABSTRACT

Satisfiability Modulo Counting (SMC) is a recently proposed general language to reason about problems integrating statistical and symbolic artificial intelligence. An SMC formula is an SAT formula in which the truth values of a few Boolean predicates are determined by model counting, or equivalently, probabilistic inference. Existing solvers optimize surrogate objectives and hence provide no formal guarantee. An exact solver is desperately needed. However, the direct integration of Satisfiability and probabilistic inference solvers results in slow performance because of many back-and-forth invocations of both solvers. We develop KOCO-SMC, a fast exact SMC solver, exploiting the fact that many similar probabilistic inferences are needed throughout SMC solving. We pre-compile the probabilistic inference part of SMC solving into probabilistic circuits, supporting efficient lower and upper-bound computation. Experiment results in several real-world applications demonstrate that our approach provides exact solutions, much better than those from approximate solvers, while is more efficient than direct integration with the current exact solvers.

## 1 INTRODUCTION

Symbolic and statistical Artificial Intelligence (AI) are two core paradigms with distinct strengths and limitations: symbolic AI, exemplified by SATisfiability (SAT) and constraint programming, excels in constraint satisfaction but cannot handle probability distributions. Statistical AI captures probabilistic uncertainty but does not guarantee to satisfy the symbolic constraints. Integrating symbolic and statistical AI remains an open field and has gained much research attention recently (Freuder & O'Sullivan, 2023; nes, 2023; neu, 2023).

As a motivating example, a manager needs to determine a set of supplier channels to ensure sufficient raw materials for good production, taking into account stochastic events such as natural disasters or policy risks. This robust supply chain design problem necessitates both symbolic reasoning to find satisfiable supplier channels and statistical inference to ensure the suppliers are robust to stochastic natural disasters. Slightly modified problem formulations can be found in many real-world applications including vehicle routing (Toth & Vigo, 2002), internet resilience (Israeli & Wood, 2002), social influence maximization (Kempe et al., 2005), energy security (Almeida et al., 2019), etc.

The recently proposed Satisfiability Modulo Counting (SMC) (Fredrikson & Jha, 2014; Li et al., 2024) provides a general language to reason about problems integrating statistical and symbolic AI, including the aforementioned supply chain design problem. Specifically, an SMC formula is an SAT formula in which the truth values of a few Boolean predicates are determined by model counting, which calculates the number of distinct variable assignments so that the SAT formula evaluates to true. The statistical reasoning is formulated as the model counting because inference over marginal probability distribution can be cast as weighted model counting problems (Chavira & Darwiche, 2008). Solving SMC problems poses significant challenges since they are $NP^{PP}$-complete (Park & Darwiche, 2004).

Several approximate SMC solvers have been proposed. Approximate solvers based on Sample Average Approximation (Kleywegt et al., 2002) were the most widely implemented, which used sample mean to estimate the model counting. Another approximate solver, XOR-SMC (Li et al., 2024), offers a constant approximation guarantee, by using the XOR-sampling to estimate the counting.

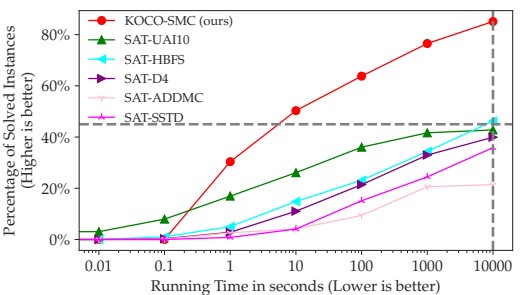 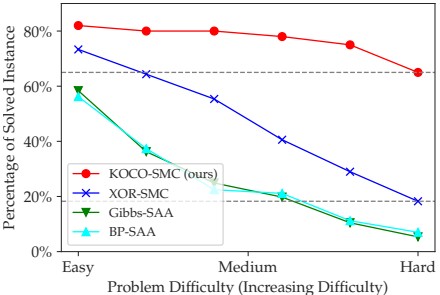

Figure 1: **(Left)** Our KOCO-SMC uses less time than all the exact baselines on the synthesized SMC dataset. Our KOCO-SMC takes 10 seconds to solve roughly 45% of SMC instances while baselines need 3 hours. **(Right)** The performance changes as the SMC problem becomes more challenging. When the SMC problem is harder to solve, approximate solvers constantly produce incorrect solutions. The performance decline of KOCO-SMC is slower than the rest.

Yet the solution quality of both methods lacks a formal guarantee, as the solution could still violate a fraction of the constraints. Current exact SMC solvers directly combine an SAT solver for satisfiability with a model counting solver for statistical inference. Specifically, the SAT solver first gives a feasible variable assignment for the Satisfiability problem, which is then checked against statistical inference constraints by a model counting solver. If the assignment fails to meet all constraints, the process restarts with a new variable assignment. This method results in an excessive number of back-and-forth invocations of SAT and model counter. Particularly for unsatisfiable problems, these exact SMC solvers enumerate all possible solutions before confirming unsatisfiability and thus are extremely slow.

We introduce KOCO-SMC, an exact and efficient SMC solver, mitigating the extreme slowness typically encountered in unsatisfiable SMC problems. KOCO-SMC saves time by detecting the conflict early and pruning the branch of the variable assignment tree. The core idea involves tracking the upper and lower bounds of the probability inside probabilistic constraints during variable assignments. If these bounds violate the satisfaction condition—such as when the upper bound of the probability falls below the required minimum. These conflicts arising from probabilistic constraints are recorded as "learned Boolean clauses" and appended to the Boolean part, preventing the same conflict from occurring in future iterations. Furthermore, integrating knowledge compilation from discrete functions into probabilistic circuits allows for rapid, repetitive updates of bounds. Consequently, our KOCO-SMC approach greatly improves the efficiency of SMC solving.

In experiments, we evaluate several approximate and exact solvers on 1350 SMC problems from the UAI Competition benchmark. Figure 1 shows the comparison with state-of-the-art solvers. Compared with exact solvers, KOCO-SMC scales the best. Our KOCO-SMC solves 85% of SMC problems in 3 hours while other exact solvers can only solve 45%, and KOCO-SMC needs 10 seconds to solve those 45% of SMC instances. Compared with those approximate solvers, KOCO-SMC reliably delivers higher quality solutions within the time limit. KOCO-SMC solves 40% more problems for hard SMC problems, whereas approximate solvers consistently produce infeasible solutions. We also demonstrate the process of formulating real-world problems, supply network design and package delivery, into the SMC formulation, and highlight the strong capability of our solver in addressing these problems.

To summarize, our main contributions are: (1) We propose KOCO-SMC, an efficient exact solver for SMC problems, integrating probabilistic circuits for effective conflict detection. (2) Experiments on synthetic datasets illustrate KOCO-SMC's superior performance compared to state-of-the-art approximate and exact baselines in both solution quality and time efficiency. (3) The case study demonstrates the potential and effectiveness of applying KOCO-SMC to real-world problems.

## 2 PRELIMINARIES

**Satisfiability Problems and Its Solver.** Satisfiability (SAT) determines whether there exists an assignment of truth values to Boolean variables that makes the entire logical formula true. Numerous

SAT solvers (Dudek et al., 2020b; Marques-Silva & Sakallah, 1999; Eén & Sörensson, 2003; Hamadi et al., 2010) have demonstrated great performance in various applications.

Conflict-Driven Clause Learning (CDCL) (Silva & Sakallah, 1996) is a modern SAT-solving algorithm that has been widely applied. The process begins by making decisions to assign values to variables and propagating the consequences of these assignments. If a conflict is encountered (i.e., a clause is unsatisfied), the solver performs conflict analysis to learn a new clause that prevents the same conflict in the future. The solver then backtracks to an earlier decision point, and the process continues. Through clause learning and backtracking, CDCL improves efficiency and increases the chances of finding a solution or proving unsatisfiability.

**Probabilistic Inference and Model Counting.** Probabilistic inference encompasses various tasks, such as calculating conditional probability, marginal probability, maximum a posteriori probability (MAP), and marginal MAP (MMAP). Each of them is essential in fields like machine learning, data analysis, and decision-making processes. Model counting calculates the number of satisfying assignments (models) for a given logical formula, and is closely related to probabilistic inference. In many scenarios, especially in discrete probabilistic models, computing probabilities can be translated to model counting, by counting the number of ways certain events or configurations can occur.

**Problem Definition for Satisfiability Modulo Counting** Satisfiability Modulo Counting (SMC) is a recently proposed extension of SAT (Fredrikson & Jha, 2014; Li et al., 2024), which incorporates constraints that involve model counting. Recognizing the intrinsic connection between probabilistic inference and model counting, SMC adaptively captures the satisfiability problem in scenarios involving uncertainty.

We use lower-case letters for random variables (i.e., $x$, $y$, $z$, and $b$) and use bold symbols (i.e., $\mathbf{x}$, $\mathbf{y}$, $\mathbf{z}$ and $\mathbf{b}$) as vectors of Boolean variables, e.g., $\mathbf{x} = (x_1, \ldots, x_n)$. Each variable $x_i$ takes binary values in $\{\texttt{False}, \texttt{True}\}$. Given a formula $\phi$ for Boolean constraints and weighted functions $\{f_i\}_{i=1}^K$, and $\{g_i\}_{i=1}^K$ for the discrete probability distributions, the SMC problem is to determine if the following formula is satisfiable over random variables $\mathbf{x} = (x_1, x_2, \ldots, x_n), \mathbf{y}_i = (y_1, y_2, \ldots, y_n), \mathbf{z}_i = (z_1, z_2, \ldots, z_n)$ and $\mathbf{b} = (b_1, b_2, \ldots, b_L)$:

$$\phi(\mathbf{x}, \mathbf{b}), \quad \text{where } b_i \Leftrightarrow \sum_{\mathbf{y}_i} f_i(\mathbf{x}, \mathbf{y}_i) \geq q_i \ \text{ or } \ b_i \Leftrightarrow \sum_{\mathbf{y}_i} f_i(\mathbf{x}, \mathbf{y}_i) \geq \sum_{\mathbf{z}_i} g_i(\mathbf{x}, \mathbf{z}_i). \tag{1}$$

Each $f_i$ (or $g_i$) is an unnormalized discrete probability function over Boolean variables in $\mathbf{x}$ and $\mathbf{y}_i$ (respectively, $\mathbf{x}$ and $\mathbf{z}_i$). Hence, $\sum f_i$ and $\sum g_i$ compute the marginal probabilities, with $\mathbf{y}_i$ and $\mathbf{z}_i$ as latent variables that are marginalized out. Thus, only $\mathbf{x}$ and $\mathbf{b}$ are decision variables. Each $b_i$ is referred to as a *Probabilistic Predicate*, which is evaluated as true if and only if the inequality over the marginalized probability is satisfied. Each probabilistic constraint is in the form of either (1) the marginal or joint probability surpassing a given threshold $q_i$, or (2) one marginal joint probability being greater than another. Note that the biconditional "$\Leftrightarrow$" can be generalized to "$\Rightarrow$" or "$\Leftarrow$", inequality "$\geq$" case in the above definition can be generalized to "$=, >$" cases and the reversed direction inequality "$\leq, <$" cases.

In summary, this SMC formulation provides a general language to reason about problems integrating symbolic and statistical constraints. Specifically, the symbolic constraint is characterized by a Boolean satisfiability formula $\phi$. The statistical constraint is captured by constraints involving the weighted model counting term $\sum f_i$.

**Probabilistic Circuits** (PCs) are a broad category of probabilistic models known for enabling a variety of exact and efficient inferences (Darwiche, 2002; 1999; Poon & Domingos, 2011; Rahman et al., 2014; Kisa et al., 2014; Dechter & Mateescu, 2007; Vergari et al., 2020; Peharz et al., 2020). Formally, PC is a computational graph encoding a probability distribution $P(\mathbf{x})$ over a set of random variables $\mathbf{x}$. The graph is composed of leaf nodes, product nodes, and sum nodes. Each node $v$ represents a probability distribution over certain random variables. Figure 2(c) gives an example PC over four variables. A leaf node $u$ encodes a tractable probability distribution $P_u(x_i)$ over a single random variable $x_i$, such as Gaussian or Bernoulli distributions. A product node $u$ defines a factorized distribution $P_u(\mathbf{x}) = \prod_{v \in ch(u)} P_v(\mathbf{x})$ where $ch(u)$ denotes the children nodes of $u$. A sum node $u$ represents a mixture distribution $P_u(\mathbf{x}) = \sum_{v \in ch(u)} w_v P_v(\mathbf{x})$, where $w_v$ represent the normalization weights of child node $v$. The root node $r$ in the graph has no parent node. A probabilistic circuit is a model of its root node distribution.

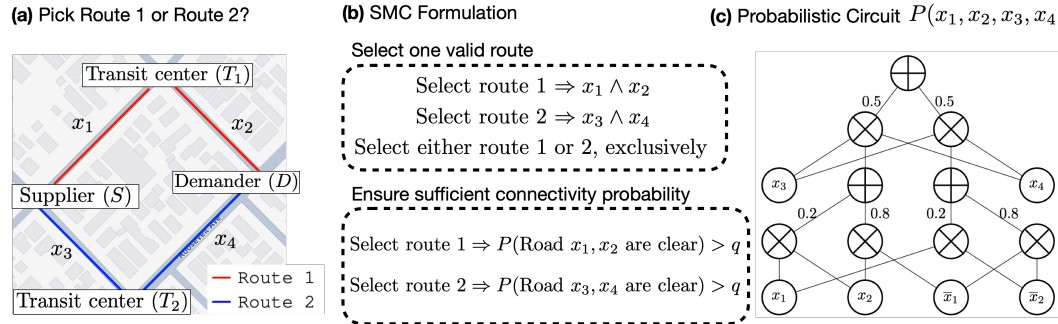

Figure 2: Example of formulating the robust supply chain problem into SMC. **(a)** shows a road map containing 4 locations and the road between them. The connectivity of each road is denoted by a random variable $x_i$, where $x_i = \texttt{True}$ indicates the corresponding road is well connected. **(b)** Model the supply routine planning as an SMC problem. **(c)** the probability of every connectivity situation, represented as the Probabilistic Circuit data structure. Each $x_i$ or $\overline{x}_i$ node denotes a leaf node that encodes a Bernoulli distribution $P(x_i = \texttt{True}) = 1$ or $P(x_i = \texttt{False}) = 1$. $\oplus$ and $\otimes$ represent the sum and product nodes respectively. The values next to the edges are weights for summation nodes.

Probabilistic circuits with specific structural properties enable efficient probability inferences, scaling polynomially with circuit size. For example, partition functions and marginal probabilities are computed efficiently due to decomposability and smoothness, MAP requires determinism for maximization, and MMAP further requires Q-determinism Choi et al. (2022).

The process of transforming a probability distribution into a probabilistic circuit with a specific structure is referred to as *knowledge compilation*. Several knowledge compilers, such as ACE (Darwiche & Marquis, 2002), C2D (Darwiche, 2004), and D4 (Lagniez & Marquis, 2017), have been developed to convert discrete distributions into tractable PCs for various probabilistic inference tasks.

## 3 METHODOLOGY

### 3.1 MOTIVATION

We use supply chain design as a case study for the SMC problem to highlight the limitations of current approximate and exact SMC solvers. As shown in Figure 2(a), the task is to deliver raw materials from suppliers to demanders on a road map. We slightly abuse the notation for $x_i$, where $i = 1, 2, 3, 4$. In the Boolean formula $\phi$, $x_i = \texttt{True}$ represents the selection of road $x_i$ in the route. In the probabilistic constraint, however, $x_i = \texttt{True}$ indicates that road $x_i$ is clear. The choice is between route 1 ($x_1 = x_2 = \texttt{True}$) and route 2 ($x_3 = x_4 = \texttt{True}$). This choice is captured by the Boolean variables $b_1$ and $b_2$, where $b_1 = \texttt{True}$ (and similarly $b_2 = \texttt{True}$) indicates the selection of route 1 (or route 2, respectively).

Various random events, such as natural disasters and car accidents, may affect road connectivity. Such stochasticity is formulated as a joint probability distribution over all roads $P(x_1, x_2, x_3, x_4)$ (modeled as a probabilistic circuit in Figure 2(c)). The probability of $x_1$ and $x_2$ being well connected is the marginal probability $P(x_1, x_2 \text{ are clear}) = \sum_{x_3, x_4} P(x_1 = x_2 = \texttt{True}, x_3, x_4)$.

Let $q \in [0, 1]$ represent the minimum required probability of good connectivity along the route. The task of selecting a route that guarantees a sufficient probability (Figure 2(b)) can be formulated as an SMC problem:

$$\phi(\mathbf{x}, \mathbf{b}) = \underbrace{(b_1 \oplus b_2)}_{(a)} \wedge \underbrace{(b_1 \Rightarrow x_1 \wedge x_2)}_{(b)} \wedge \underbrace{(b_2 \Rightarrow x_3 \wedge x_4)}_{(c)},$$

$$\underbrace{b_1 \Rightarrow \sum_{x_3, x_4} P(x_1, x_2, x_3, x_4) \geq q}_{(e)}, \qquad \underbrace{b_2 \Rightarrow \sum_{x_1, x_2} P(x_1, x_2, x_3, x_4) \geq q}_{(f)} \qquad (2)$$

where $\oplus$ is the logical "exclusive or" operator. In part (a), the constraint ensures that only one route is selected. In part (b), the clause $(b_1 \Rightarrow x_1 \wedge x_2)$ indicates that: if route 1 is selected, both $x_1$ and $x_2$ must be assigned $\texttt{True}$, representing the edge selection. Part (c) applies a similar condition for route 2. In part (e), $\sum_{x_3, x_4} P(x_1, x_2, x_3, x_4) = P(x_1, x_2)$ marginalizes out $x_3$ and $x_4$, representing the probability of route 1's condition under random factors. Part (f) is analogous to part (e).

An existing SMC solver (like the SAT-* solver in our experiments) finds the routine in the following manner. If we set $q = 0.5$,

1. It first uses an SAT solver to solve the Boolean SAT problem $\phi(\mathbf{x}, \mathbf{b})$ and proposes a solution, e.g., $x_1 = x_2 = b_1 = \texttt{True}, x_3 = x_4 = b_2 = \texttt{False}$ (indicating route 1).

2. Then, it infers the marginal probability $\sum_{x_3, x_4} P(x_1 = x_2 = \texttt{True}, x_3, x_4) = 0.1 < 0.5$, and find it violates the probabilistic constraint.

3. Setting $x_1 = x_2 = \texttt{True}$ causes a conflict, so we add clause $(\overline{x}_1 \vee \overline{x}_2)$ to $\phi$ to avoid the same conflict. We then return to step 1 and propose a new assignment.

In this process, the SAT solver and probability inference are sequentially dependent, each waiting for the other to finish, which results in time waste. In contrast, our KOCO-SMC immediately detects a conflict upon the partial assignment $x_1 = \texttt{True}$, saving time by avoiding further assignments to the remaining variables. Although $x_2$ is still unassigned, the highest possible probability value is below 0.5, i.e., $\max_{x_2} \sum_{x_3, x_4} P(x_1 = \texttt{True}, x_2, x_3, x_4) = 0.1 < 0.5$. This should trigger an immediate conflict instead of waiting for the SAT solver to assign $x_2$. Therefore, KOCO-SMC can solve SMC problems more efficiently.

## 3.2 MAIN PIPELINE OF KOCO-SMC

This section presents the proposed KOCO-SMC approach for solving SMC problems both exactly and efficiently. KOCO-SMC follows the Conflict-Driven Clause Learning (CDCL) framework (Silva & Sakallah, 1996; Eén & Sörensson, 2003) (Algorithm 1 in the Appendix), which comprises four key components: variable assignment, propagation, conflict clause learning, and backtracking. With the inclusion of probabilistic constraints in SMC problems, KOCO-SMC is further tailored to efficiently address these challenges.

**Pre-compilation.** Before SMC problem solving, a knowledge compilation step compiles all distributions in probabilistic constraints into smooth, decomposable, and Q-deterministic probabilistic circuits. An example is provided in Figure 2(c).

**Propagation.** In the CDCL algorithm, unit propagation is used to propagate new variable assignments across clauses. This process can create additional variable assignments or detect conflicts. For example, if we have the clauses $(x_1 \vee \neg x_2)$ and $(x_2 \vee x_3)$, and we assign $x_1 = \texttt{False}$, unit propagation would force $x_2 = \texttt{False}$, leading to further propagation $x_3 = \texttt{True}$. This significantly accelerates SAT solving. However, this procedure is specifically designed for Boolean clauses. How to incorporate probabilistic constraints into the propagation process, extract useful information from current variable assignments, and effectively detect conflicts remains an open problem. We propose the Upper-Lower Watch (ULW) approach, an efficient propagation method for probabilistic constraints that leverages the power of probabilistic circuits. By utilizing modern knowledge compilers, most common probability distributions can be compiled into tractable probabilistic circuits, making our approach broadly applicable.

**Conflicts Clause Learning.** In the CDCL algorithm, once a conflict is detected within a Boolean clause, there are existing techniques to add a learned clause to the original Boolean formula, preventing the same conflict from occurring in the future. When a conflict arises in a probabilistic constraint, KOCO-SMC generates a learned conflict clause by negating the current variable assignments involved in the constraint and connecting them with logical OR. For example, a conflict in $\sum_y P(x_1 = \texttt{True}, x_2 = \texttt{False}, y) > Q$ will produce the clause $(\neg x_1 \vee x_2)$, preventing the assignment $x_1 = \texttt{True}, x_2 = \texttt{False}$ from being used in future iterations.

## 3.3 Upper Lower Watch for Tracking Probabilistic Constraints

The satisfaction or conflict of a probabilistic constraint is determined by the involved marginal probability. By maintaining a range of the marginal probability and refining it with each new variable assignment, we can detect satisfiability or conflict early when the range significantly deviates from the threshold. Determining the range of a marginal probability is solving the following problem:

$$\max_{\mathbf{x}_{\text{remain}}} \sum_{\mathbf{y}} P(\mathbf{x}_{\text{assigned}}, \mathbf{x}_{\text{remain}}, \mathbf{y}), \quad \min_{\mathbf{x}_{\text{remain}}} \sum_{\mathbf{y}} P(\mathbf{x}_{\text{assigned}}, \mathbf{x}_{\text{remain}}, \mathbf{y}) \qquad \text{(Marginal MAP)}$$

where $\mathbf{x}_{\text{assigned}}$ denotes the assigned variables, $\mathbf{x}_{\text{remain}}$ represents the unassigned variables, and $\mathbf{y}$ are the marginalized-out latent variables. This problem is known as the Marginal MAP (MMAP) inference task. As discussed in the preliminaries, probabilistic circuits with specific structural properties can efficiently compute MMAP queries. The maximum is referred to as the "upper bound," and the minimum as the "lower bound." Our Upper-Lower Watch (ULW) algorithm monitors both values.

By leveraging the pre-compilation step, we can model probabilistic distributions using Q-deterministic, decomposable, and smooth probabilistic circuits. The computation of upper and lower bounds is then reduced to performing MMAP inference on these circuits. More specifically, each node $v$ in the probabilistic circuit represents a distribution $P_v$ over a subset of variables. The ULW algorithm associates each node with an upper bound $UB(v)$ and a lower bound $LB(v)$ for the marginal probability of the sub-distribution $P_v$ under the current assignment. Therefore, the $UB$ and $LB$ at the root node provide the exact upper and lower bounds we seek.

Once a new variable is assigned, the bounds associated with each node can be updated in a bottom-up manner. The update scheme for the leaf nodes is as follows:

- For a leaf node $v$ over variable $x$ (the variable to be decided), update $UB(v) = \max\{P_v(x = \text{True}), P_v(x = \text{False})\}$ and $LB(v) = \min\{P_v(x = \text{True}), P_v(x = \text{False})\}$ if $x$ is not assigned. Otherwise, when variable $x$ is assigned to $val$, $UB(v) = LB(v) = P_v(x = val)$.

- For a leaf node $v$ over $y$ (the variable to be marginalized), update $UB(v) = LB(v) = 1$.

We use $ch(v)$ to denote the set of children nodes of $v$. The intermediate nodes, product nodes or sum nodes, can be updated by:

- For a product node $p$, update $UB(p) = \prod_{u \in ch(p)} UB(u)$, and $LB(p) = \prod_{u \in ch(p)} LB(u)$.

- For a sum node $s$, update $UB(s) = \max_{u \in ch(s)} w_u UB(u)$, and $LB(s) = \min_{u \in ch(s)} w_u LB(u)$.

This updating mechanism only updates paths from the updated leaf nodes to the root, which ensures its efficacy. The correctness is guaranteed by Q-determinism, smoothness and decomposability property of the probabilistic circuit. A former justification is in Lemma 1, see proof in the Appendix C. This bound-updating scheme forms the propagation process.

The bounds at the root node represent the bounds for the marginal probability in the constraint. Conflict detection then becomes straightforward. For example, consider the constraint $b \Leftrightarrow \sum_{\mathbf{y}} P(\mathbf{x}, \mathbf{y}) \geq q$, where $P(\mathbf{x}, \mathbf{y})$ has been compiled into a probabilistic circuit rooted at node $root$. If $UB(root) < q$ or $LB(root) \geq q$, we can safely conclude that there is a conflict or that the constraint is certainly satisfied, respectively.

In addition, frequent variable assignments can empirically cause excessive delays due to the need for constant bound updates. Inspired by the Two-Literal Watch technique (Marques-Silva & Sakallah, 1999), where the propagation reaches a Boolean clause only when two specific literals are newly assigned—regardless of the number of literals in the clause—we apply a similar strategy to probabilistic constraints. We define two watched variables for each probabilistic constraint: one decision variable and the probabilistic predicate. For instance, in $b_1 \Leftrightarrow \sum_{y_1, y_2} P(x_1, x_2, y_1, y_2)$, we designate $b_1$ and either $x_1$ or $x_2$ as the watched variables. The upper and lower bounds of $\sum_{y_1, y_2} P(x_1, x_2, y_1, y_2)$ are updated only when one of the watched variables is assigned, and this process continues until no unassigned variables remain.

**Assumption 1** (Q-Deterministic, Smooth and Decomposable (Choi et al., 2022)). *A smooth probabilistic circuit when all children of every sum node share identical sets of variables; A probabilistic*

*circuit is decomposable if the children of every product node have disjoint sets of variables; A probabilistic circuit is Q-deterministic when a partial assignment of all query (decision) variables ensures that no more than one child of the sum node produces a non-zero output. Smoothness, decomposability, and Q-determinism enable tractable computation of any MMAP query.*

**Lemma 1.** *Suppose $P(\mathbf{x}, \mathbf{y})$ is a Q-deterministic and decomposable probabilistic circuit defined over Boolean variables $\mathbf{x} = (x_1, \ldots, x_N)$ and $\mathbf{y} = (y_1, \ldots, y_M)$. Our ULW algorithm finds exact bounds.*

*Sketch of proof.* The result is obtained by applying the theoretical properties of Q-deterministic, smooth, and decomposable probabilistic circuits to solve the marginal MAP problem. Please refer to Appendix C for a detailed proof. □

## 4 EXPERIMENTS

### 4.1 EXPERIMENT SETTINGS

For the experiment, we consider the satisfiability of the following SMC problem: $\phi(\mathbf{x}) \wedge \left( \sum_{\mathbf{y}} f(\mathbf{x}, \mathbf{y}) > Q \right)$ where $\phi(\mathbf{x})$ is a Boolean formula in CNF, $f$ is a (unnormalized) probability distribution, $\mathbf{x}$ denotes the set of decision variables, $\mathbf{y}$ represents variables to be marginalized, and $Q \in \mathbb{R}$ is the threshold value. We exclude the variable $\mathbf{b}$ from Equation 1 and fix the number of counting constraints to one in order to better control the properties of SMC problems. This allows us to gain clearer insights into the capabilities of our solver.

**Dataset** For $f$ in the probabilistic constraints, we used the partition function-task benchmark that appears in the Uncertainty in Artificial Intelligence (UAI) Challenge from 2010 and 2022. 50 models over binary variables are kept. The remaining models can be grouped by 6 categories *Alchemy* (1 model), *CSP* (3 models), *DBN* (6 models), *Grids* (2 models), *Promedas* (32 models), and *Segmentation* (6 models). For $\phi$ in the Boolean satisfiability, we randomly generated 9 different 3-coloring problems, in CNF, using CNFgen. The number of involved binary variables ranged from 75 to 675. The threshold value $Q$ varies according to the task, and will be detailed in the respective sections.

**Implementation of KOCO-SMC** We applied ACE (Darwiche & Marquis, 2002) as the knowledge compiler. The CDCL skeleton of KOCO-SMC is implemented on top of MiniSAT (Eén & Sörensson, 2003), for its easily extensible structure. For the ablation study, we include a version without ULW (KOCO-SMC without ULW).

**Baselines** We consider several approximate SMC solvers and exact SMC solvers. For the approximate solver, we include the Sampling Average Approximation (SAA) (Kleywegt et al., 2002)-based method. Specifically, we use Lingeling (Biere, 2017) SAT solver to enumerate solutions and estimate $\sum_{\mathbf{y}} f(\mathbf{x}_f, \mathbf{y})$ by an average over samples, which enables approximate inference of marginal probabilities. We include Gibbs Sampler (Shapiro, 2003) (Gibbs-SAA) and Belief Propagation (BP-SAA) (Fan & Yexiang, 2020). We also include XOR-SMC (Li et al., 2024), an approximated solver specifically for SMC problems.

The baseline exact solver is composed of an exact SAT solver and probabilistic inference solvers. This approach first identifies a solution to the Boolean formula and then sequentially verifies it against the probabilistic constraints. For the Boolean SAT solver, we selected Lingeling (Biere, 2017) for its superior performance. For probabilistic inference, we selected top-performing solvers from the Uncertainty in Artificial Intelligence (UAI) Competitions. Due to limited access to these solvers, we chose the UAI2010 winning solver implemented in libDAI (Mooij, 2010) (SAT-UAI10) and the solver based on the hybrid best-first branch-and-bound algorithm (SAT-HBFS) developed by Toulbar2 (Cooper et al., 2010). Although Toulbar2 was not the winner of the Partition Function or Marginal Probability tracks, it offers the necessary functionality and demonstrated strong performance in our tests. Due to the underlying connection between probabilistic inference and weighted model counting, we also include model counters from recent Model Counting competitions (Fichte et al., 2021) from 2020 to 2023: d4 solver (Lagniez & Marquis, 2017) (SAT-D4), ADDMC (Dudek et al., 2020a) (SAT-ADDMC), and SharpSAT-td (Korhonen & Järvisalo, 2023) (SAT-SSTD).

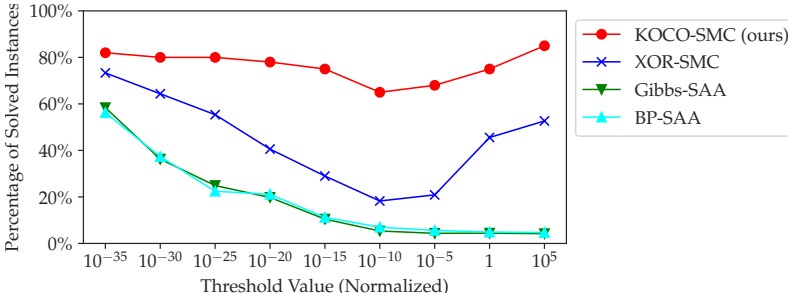

Figure 3: Comparison of KOCO-SMC with approximate solvers on solving SMC instances across varying thresholds. As the threshold increases from $10^{-35}$ to $10^{-10}$, the chance of discovering satisfying configurations decreases, leading to a drop in all solvers' performance. From $10^{-5}$ to $10^5$, most SMC problems become unsatisfiable. Higher thresholds impose more extreme conditions, allowing solvers to quickly determine unsatisfiability, resulting in improved performance. KOCO-SMC consistently outperforms others, maintaining a high percentage of solved instances.

## 4.2 RESULT ANALYSIS

**Comparison with Approximate Solvers** Our approach is compared with baselines across a total of $9 \times 50$ combinations of benchmark CNF and probabilistic models. For each combination, we use the partition function of the probabilistic model multiplied by various scalars as the varying thresholds. The scalars range from $10^{-35}$ to $10^5$, as shown on the x-axis of Figure 3. Each approximate solver runs 5 times on each problem, and if one correct solution is found, the problem is considered 'solved'. The portion of solved instances in 1 hour is shown in Figure 3.

As the threshold increases from $10^{-35}$ to $10^{-10}$, most SMC problems remain satisfiable, but the likelihood of finding a satisfying configuration decreases, causing a performance decline across all solvers. Between $10^{-5}$ and $10^5$, most problems become unsatisfiable. Higher thresholds create more extreme conditions, enabling solvers to quickly detect unsatisfiability and improve performance. Overall, KOCO-SMC demonstrates its strong performance across varying threshold levels.

**Comparison with Exact Solvers** We study the performance of different exact SMC solvers facing SMC problems with different numbers of satisfying solutions. This is accomplished by changing the value of threshold $Q$ and measuring the solving time. As the increase of $Q$, satisfying assignments become rare and finally unsatisfiable.

An illustrating result on a specific combination (*3-color-5x5.cnf* with *smokers_10.uai*) is shown in Figure 4(left). At low thresholds, all approaches quickly find a satisfying assignment; KOCO-SMC's initial time is higher than average due to the pre-compilation of the probabilistic model. As the threshold rises, satisfying assignments become rare, leading to increased time costs. Notably, after the problem shifts from satisfiable to unsatisfiable, our methods' (KOCO-SMC) time cost decreases, while others maintain high time consumption. This efficiency is due to our integrated ULW propagation, allowing early identification of unsatisfiability, while others have to enumerate all possible solutions before conclusion. In cases of extremely high thresholds, our method concludes unsatisfiability immediately.

**Runtime Evaluation** The efficiency of the KOCO-SMC can be further illustrated by the evaluation of the whole benchmark. We pick three kinds of threshold values: a critical threshold beyond which the SMC becomes UNSAT, 50% of the critical threshold, and 150% of the critical threshold. We have 1350 different SMC instances in total. Figure 4 shows the relation of the percentage of solved instances with the running time. KOCO-SMC exhibits significantly better performance.

## 4.3 CASE STUDIES

**Effectiveness of Upper-Lower Bound Watch** In Figure 5(left), ULW propagation accelerates KOCO-SMC by 10 times compared with KOCO-SMC without ULW when the threshold reaches

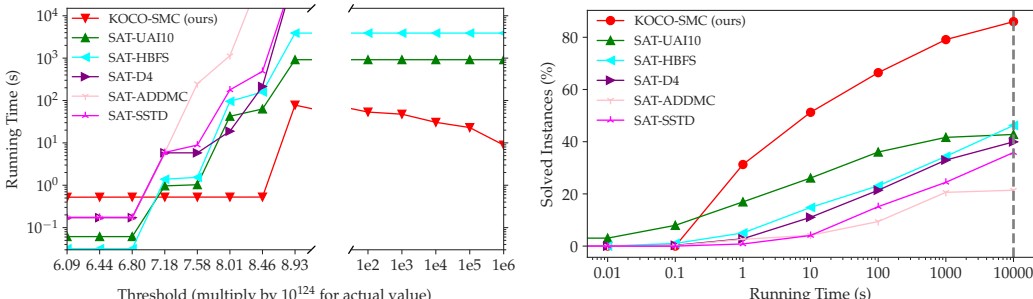

Figure 4: Comparing with exact solvers, KOCO-SMC solves $80\%$ of SMC problems in 20 minutes while others solve $40\%$ in 3 hours. **(Left)** The running time (x-axis) of experiments on a specific CNF and a Probabilistic Model with varying thresholds (y-axis). Our method typically requires significantly less time across most instances. Particularly when the threshold exceeds the critical point at which the SMC becomes UNSAT, our approach exhibits an improved performance. **(Right)** The percentage of instances solved in a given time limit.

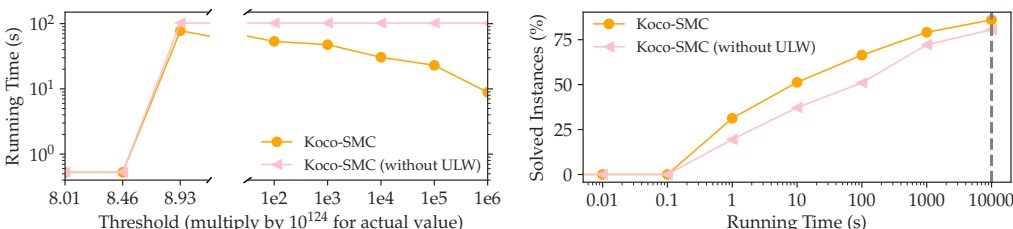

Figure 5: UWL in KOCO-SMC is shown to be the key component in accelerating SMC solving. **(Left)** The running time with varying thresholds. ULW propagation accelerates KOCO-SMC by 10 times compared with KOCO-SMC without ULW when the threshold reaches $10^{130}$. **(Right)** The percentage of instances solved in a given time limit.

$10^{132}$. Figure 5(right) further demonstrates the contribution of ULW, where KOCO-SMC is 10 times faster than KOCO-SMC without UWL for SMC problems solvable around 10 minutes.

**Application: Supply Chain Design** The objective is to develop the best trading plan for each supplier in the supply chain network, ensuring that all trades have the highest success probability and satisfy the budget constraints. In the supply chain network, each supplier is represented as a node, which purchases raw materials from upstream nodes and sells products to downstream nodes. (1) To balance manufacturing safety and budget constraints, each node purchase raw materials from exactly two upstream suppliers and sells to exactly two downstream customers. (2) Trades may be disrupted by random events such as natural disasters, car accidents, or political issues. The trading plan must ensure a minimum probability of all trades succeeding, guaranteeing resilience against disruptions.

Let $x_e \in \{\texttt{True}, \texttt{False}\}$ represent the selection of a trade between nodes connected by edge $e$, where $x_e = \texttt{True}$ if the trade is selected. Combining the requirement (1) and (2), we have the SMC formulation: $\phi(\mathbf{x}_e) \wedge (\sum_{\mathbf{x}'} P(\mathbf{x}_e, \mathbf{x}') > Q)$ where $\phi(\mathbf{x}_e)$ represents the budget constraints on the set of selected trades $\mathbf{x}_e$, $Q$ is the minimum requirement of successful probability, and $P(\cdot)$ is the probabilistic transportation model defined over all edges. The marginal probability $\sum_{\mathbf{x}'} P(\mathbf{x}_e, \mathbf{x}')$ is the probability that all selected trades are carried out successfully.

We use 4-layer supply chain networks from the bread supply chain dataset containing 44 nodes (Large) (Zokaee et al., 2017), where each layer represents a tier of suppliers. Additionally, we introduce synthetic networks with 20 nodes (Small) and 28 nodes (Medium) to improve illustration. To find the plan guaranteeing the highest success probability, we gradually increase the threshold $Q$ from 0 to 1 in increments of $1 \times 10^{-2}$, continuing until the threshold makes the SMC problem unsatisfiable. The running time for finding the best plan is shown in Figure 6 (Left), and detailed

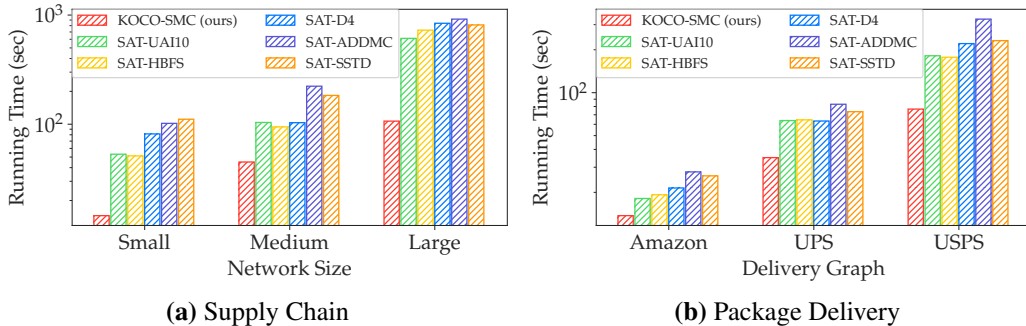

**(a)** Supply Chain          **(b)** Package Delivery

Figure 6: **(Left)** Running time of each method for identifying the best trading plan. All methods are tested on three real-world supply chain networks of different sizes. **(Right)** Running time for identifying the best delivery path. All methods are tested on three road maps of different sizes. Our KOCO-SMC finds all the best solution significantly faster.

settings are in Appendix D.4. Through a proper problem definition, KOCO-SMC demonstrates superior performance in finding the optimal plan.

**Application: Package Delivery** The task is to find a Hamiltonian path that covers major delivery locations while minimizing the chance of encountering heavy traffic (Hoong et al., 2012). The delivery locations and roads are modeled as nodes and edges, respectively. (1) The path must be Hamiltonian, passing through each node exactly once. (2) Each road segment has a probability of heavy traffic, depending on the time of travel, weather conditions, and road properties. The probability of encountering heavy traffic on any segmentation should be lower than a threshold.

Suppose there are $N$ delivery locations indexed from 1 to $N$. Let $x_{i,j} \in$ True, False, where $x_{i,j} =$ True if and only if the $j$-th location is visited in the $i$-th position of the path. Combining requirements (1) and (2), we derive the SMC formulation: $\phi(\mathbf{x}) \land (\sum_{\mathbf{e}} P(\mathbf{x}, \mathbf{e}) < Q)$, where $\mathbf{x}$ is the set of decision variables $x_{i,j}, i, j \in 1, \ldots, N$, $\mathbf{e}$ is the set of latent environmental variables, and $P(\mathbf{x}, \mathbf{e})$ represents the probability of encountering heavy traffic given the path encoded by $\mathbf{x}$ with the environmental conditions $\mathbf{e}$. The marginal probability $P(\mathbf{x}, \mathbf{e})$ is the exact likelihood of encountering heavy traffic. Finally, $Q \in \mathbb{R}$ is the threshold value. Detailed settings are in Appendix D.5.

The graph structures used in our experiments are based on cropped regions from Google Maps. We consider three sets of delivery locations: 8 Amazon Lockers, 10 UPS Stores, and 6 USPS Stores. The three maps we examine are: Amazon Lockers only (Amazon), Amazon Lockers plus UPS Stores (UPS), and UPS graph with the addition of 6 USPS Stores (USPS). These graphs consist of 8, 18, and 24 nodes, respectively. The traffic condition probability is modeled by the Bayesian network from Los Angeles traffic data (West, 2020). We gradually decrease the threshold $Q$ from 1 to 0 in increments of $10^{-2}$, continuing until the threshold makes the SMC problem unsatisfiable. The running time for finding the best plan is shown in Figure 6 (Right). KOCO-SMC can efficiently discover an optimal plan with this proper SMC problem formulation.

## 5    CONCLUSION

We have introduced KOCO-SMC, an innovative approach for solving Satisfiability Modulo Counting (SMC) problems exactly. Our method is distinct from existing approaches, which typically combine SAT solvers with model counters. Instead, we introduce an early conflict detection mechanism by comparing the upper and lower bounds of probabilistic inferences. Through knowledge compilation, our proposed Upper Lower Watch algorithm enables efficient tracking of both bounds. Our experiments on synthetic benchmark problems demonstrate that our approach achieves superior solution quality compared to approximate solvers and significantly outperforms existing exact solvers in terms of efficiency. The real-world application highlights the potential of solving practical problems due to the high generalizability of the SMC formulation.

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

**Availability of KOCO-SMC and Dataset** Please find our code repository at:

```
https://anonymous.4open.science/r/anonym_koco_smc-61FD/
```

It contains 1) the implementation of our KOCO-SMC method, 2) the list of datasets, and 3) the implementation of several baselines.

**Limitations** Our KOCO-SMC requires all probability distributions to be compiled into Q-deterministic, decomposable, and smooth probability circuits. However, due to the limitations of knowledge compilers, compiling a complex distribution can (1) take too much time or space, (2) introduce arithmetic errors, and (3) many knowledge compilers don't support the structural requirements. The knowledge compiler we used in experiments doesn't guarantee to generate the Q-deterministic circuit. There is a gap between our theoretical analysis and the experimental results.

A relaxed version of ULW could be proposed , but it doesn't guarantee tight upper and lower bounds. As a result, the loose bounds could potentially slow down the detection of satisfaction or conflict, representing a trade-off between fast-solving and structural requirements. We will explore it in the future.

**Broader Impact** Satisfiability Modulo Counting (SMC) extends traditional Boolean satisfiability by incorporating constraints that involve probability inference (model counting). This extension allows for solving complex problems where both logical and probabilistic constraints must be satisfied. SMC has significant applications in supply chain design, shelter allocation, scheduling problems, and many others in Operation Research. For example, in scheduling problems, SMC can ensure that selected schedules meet probabilistic events, while in shelter allocation, it can verify that the accessibility under random disasters is above a specified threshold.

## A EXTENDED RELATED WORKS

Satisfiability Modulo Counting (SMC) was first introduced by Fredrikson & Jha (2014), aiming to integrate logical satisfiability with probabilistic constraints to address complex decision-making problems. Due to the novelty of this formulation, only a few specialized solvers have been proposed. For instance, Li et al. (2024) introduced an approximate solver that employs XOR constraints to relax SMC problems, enabling more tractable computations.

Other approaches often focus on specific subclasses of SMC problems. A representative example is Marginal MAP inference, which maximizes a marginal probability over query variables. Its formulation is identical to SMC problems with a single probabilistic constraint. Choi et al. (2022) developed exact solvers for Marginal MAP by transforming probabilistic circuits. Marinescu et al. (2014) uses AND-OR search approach to solve the MMAP problem.

Another related domain is Stochastic Satisfiability (SSAT), introduced by Papadimitriou (1985). SSAT can further solve SMC problems with a Boolean constraint and a single probabilistic constraint by combining Boolean SAT with probabilistic quantifiers. Subsequent research has advanced SSAT solvers (Lee et al., 2017; 2018; Fan & Jiang, 2023). However, despite their power, these solvers cannot generalize to SMC problems with multiple probabilistic constraints.

The principle idea of our Upper-Lower Bound algorithm is to use the upper and lower bounds of probabilities to avoid unnecessary search. Similar ideas can be found in branch-and-bound algorithm Boyd & Mattingley (2007), which maintains upper and lower bounds of the objective function to prune suboptimal branches of the search tree, and in alpha-beta pruning Knuth & Moore (1975), which uses bounds on the evaluation of game states to eliminate branches in the game tree that cannot affect the final decision.

## B EXTENDED METHODOLOGY

### B.1 PROBABILISTIC INFERENCE THROUGH PROBABILISTIC CIRCUITS

The inference of probabilities from a probability circuit can be very efficient. Figure 7 shows a decomposable, smooth, and deterministic probability circuit. For $P(x_1 = x_3 = x_4 = \text{T}, x_2 = \text{F})$, set the value of nodes $x_1$, $x_3$, and $x_4$ to 1, and $\overline{x}_1$ to 0. Set nodes $x_2$ and $\overline{x}_2$ oppositely since $x_2 = \text{F}$.

Evaluate the value at the root node as the probability, which is 0.1. This inference doesn't require any special property of the probabilistic circuit.

It is different for the marginal probability. We need the circuit to be decomposable and smooth to ensure efficacy. For $P(x_3 = x_4 = \texttt{T})$, set nodes $x_3$ and $x_4$ to 1 as they are assigned $\texttt{T}$. For the marginalized-out variables $x_1$ and $x_2$, set all related nodes, e.g., $x_1$ and $\overline{x}_1$, to 1. Evaluate the value at the root node, which should be 1.0.

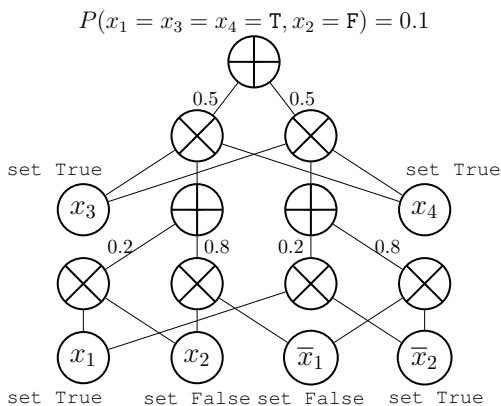

**(a)** compute probability $P(x_1 = x_3 = x_4 = \texttt{T}, x_2 = \texttt{F})$.

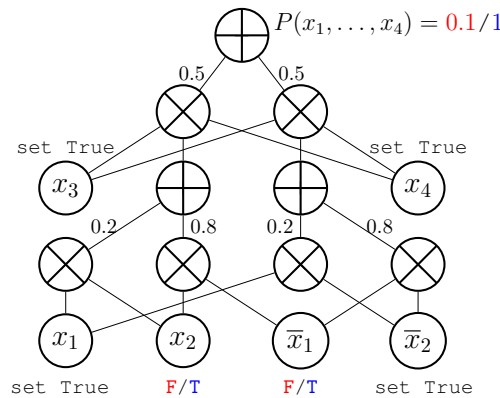

**(b)** compute marginal probability $P(x_3 = x_4 = \texttt{T})$.

Figure 7: To infer the probability $P(x_1 = x_3 = x_4 = \texttt{T}, x_2 = \texttt{F})$, set the value of nodes $x_1$, $x_3$, and $x_4$ to 1, and $\overline{x}_1$ to 0. Set values for $x_2$ and $\overline{x}_2$ oppositely since $x_2 = \texttt{F}$. The value assignment is shown in red, and the circuit evaluates to the probability 0.1. Similarly, to infer the marginal probability $P(x_3 = x_4 = \texttt{T})$, set nodes $x_3$ and $x_4$ to 1. For the marginalized-out variables $x_1$ and $x_2$, set all related nodes to 1. The value assignment is shown in blue, and the circuit evaluates to the marginal probability 1.0.

## B.2 KOCO-SMC MAIN PIPELINE

Classical SAT solvers like MiniSAT Eén & Sörensson (2003) have achieved high performance in real-world applications. We implement our method based on their MiniSAT version 2.2.0[1] The decision and backtrack steps are primarily from their implementation, but our propagation and conflict clause learning steps differ. The pseudocode is shown in Algorithm 1.

**Pre-Compilation** The tractable probabilistic circuits are constructed from the discrete probability distributions in the form of Bayesian networks or Markov Random Fields. The pipeline is introduced in Darwiche (2002) (Fig. 8) that compiles a distribution into a Boolean formula augmented

---

[1]MiniSAT: https://github.com/niklasso/minisat

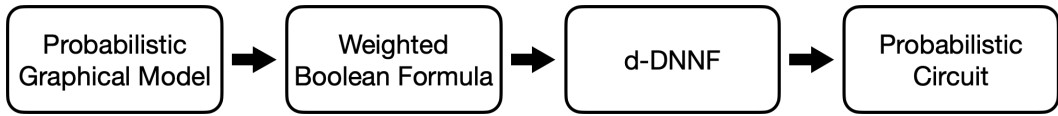

Figure 8: The process of constructing probabilistic circuits from probabilistic graphical models by ACE.

with literal weights, and is further compiled into a tractable Boolean circuit—characterized by its determinism, decomposability, and smoothness. From this circuit, one derives a tractable probabilistic circuit. We use the knowledge compilation tool: ACE[2] using their default *compile* script.

**Decision**    To quickly identify a satisfying solution, the decision is made according to some decision heuristics. KOCO-SMC utilizes Variable State Independent Decaying Sum (VSIDS) Dudek et al. (2020b) as the decision heuristic. Generally, each variable assignment is associated with a priority score. A higher score indicates a higher priority of being decided. During SMC-solving, once current variable assignments make the SMC problem unsatisfiable (also referred to as a $conflict$). The priority of those assignments will all decrease. All priority scores are then reduced by multiplying with a constant less than one. A variable's priority score is dynamically updated to reflect its recent involvement in conflicts.

**Propagation**    The detailed implementation of propagation involving probabilistic constraints is shown in Algorithm 2, which corresponds to lines 4-8 of the Algorithm 1 in the main text. The explanation is as follows.

Pick one new variable assignment ("new" refers to "hasn't been propagated"): variable $x$ assigned with value $val \in \{True, False\}$. The variable $x$ is associated with a *watcher list*, denoted by $watcher(x)$, where each element is either a Boolean clause or a probabilistic constraint involving $x$. Once $x$ is assigned with a value, only elements in $watcher(x)$ should verify its satisfiability under the current variable assignment. This mechanism is invented by Eén & Sörensson (2003). It can avoid examining all constraints involving $x$ and can improve efficacy, especially in problems with lots of constraints.

Consider a Boolean clause or probabilistic constraint $C$ in $watcher(x)$. If $C$ is a Boolean clause, we simply run the unit propagation. Otherwise, for each probabilistic circuit $c_r$ in $C$, update the upper and lower bounds of each marginal probability encoded by $c_r$ with current variable assignments. For example, a probabilistic constraint in the form of $b \Leftrightarrow \sum_y P(x,y) > Q$ contains one circuit encoding $P(x,y)$, we update the upper and lower bounds of $\sum_y P(x,y)$ with the current assignment of $x$. The detailed updating rule is specified in Section 3.3 of the main text. Then we can check the satisfiability with updated bounds, e.g., comparing the bounds of $\sum_y P(x,y)$ with threshold $Q$ in the abovementioned example. If the comparison produces a conflict, e.g., the upper bound of $\sum_y P(x,y)$ is already below $Q$, then Algorithm 2 returns a conflict with the reference to current constraint as the reason (specified in line 10-11). Otherwise, we pick a new unassigned variable to watch, i.e., put the current constraint to the watcher list of another variable. Noted that we don't explicitly "remove" a satisfied probabilistic constraint as in MiniSAT to simplify the backtracking.

**Conflict Clause Learning**    The clause learning step in line 13 of Algorithm 1 can be explained using the following example. Suppose the conflict is caused by a probabilistic constraint $C$, and the assigned variables in $C$ are $x_1 = True$ and $x_2 = False$. Then the cause of conflict can be seen as $(\overline{x}_1 \lor x_2)$ (corresponds to line 2), which is exactly a Boolean clause in a CNF formula. Then we can utilize an experimentally effective method for Boolean SAT problems based on the First Unique Implication Point heuristic. We will not give a detailed definition here, please refer to Eén & Sörensson (2003) for detailed implementation.

---

[2]ACE: http://reasoning.cs.ucla.edu/ace

---

**Algorithm 1** Solving Satisfiability Modulo Counting Exactly with Probabilistic Circuits.

---

**Input:** Boolean Formula $\phi$, Probabilistic Constraints $\{C_i\}_{i=1}^K$.
**Output:** Satisfiability and variable assignment.
1: Knowledge compilation ($\{C_i\}_{i=1}^K$) ▷ preparation
2: **loop**
3:     Decide to assign variable $x$ to $val \in \{\mathtt{T}, \mathtt{F}\}$. ▷ decision
4:     **for** each probabilistic constraint $C$ in $\{C_i\}_{i=1}^K$ **do**. ▷ propagation
5:         Update bounds of each circuit in $C$ with $v = val$.
6:         Detect conflict by comparing bounds.
7:     **for** each Boolean clause $C'$ in $\phi$ **do**
8:         Propagate $x = val$ to Boolean clause $C'$.
9:     **if** no conflict detected **then**
10:         **if** all variables assigned **then**
11:             **return** Satisfiable, Variable assignments
12:     **else**
13:         Propose a learned clause $C_l$. ▷ clause learning
14:         $\phi \leftarrow \phi \cup \{C_l\}$
15:         **if** no variable assigned **then**
16:             **return** Unsatisfiable, no assignment
17:         **else**
18:             undoing assignments until the reason no longer holds. ▷ Backtrack

---

**Algorithm 2** Propagation

---

**Input:** The set of new assignments $S$.
**Output:** Conflict detection result
1: **while** $S$ not empty **do**
2:     $x, val \leftarrow S.pop()$ ▷ variable $x$ is assigned with value $val$
3:     **for** $C \in watcher(x)$ **do**
4:         **if** $C$ is a Boolean clause **then**
5:             Unit propagation
6:         **if** $C$ is a probabilistic constraint **then**
7:             **for** $c_r \in circuits(C)$ **do**
8:                 update bounds of the marginal probability
9:                 encoded by $c_r$
10:         **if** $C$ is unsatisfied **then**
11:             **return** CONFLICT, $C$
12:         **if** $C$ has another unassigned variable $x'$ **then**
13:             Add $C$ to $watcher(x')$
14:             Remove $C$ from $watcher(x)$
15:     **return** NO CONFLICT, No conflict reason

---

## C   Proof of Lemma 1

**Definition 1.** *Denote assigned variables in* $\mathbf{x}$ *as* $\mathbf{x}_e$ *and those not assigned as* $\mathbf{x}_h$. *The upper and lower bounds of the marginal probability with the partial variable assignment are* $\max_{\mathbf{x}_h} \sum_{\mathbf{y}} P(\mathbf{x}_e, \mathbf{x}_h, \mathbf{y})$ *and* $\min_{\mathbf{x}_h} \sum_{\mathbf{y}} P(\mathbf{x}_e, \mathbf{x}_h, \mathbf{y})$.

*Proof.* Inferring the strict upper and lower bounds is known as a Marginal Maximum a Posterior (MMAP) problem. An MMAP problem can be inferred from a Q-deterministic, smooth and decomposable probabilistic circuit encoding $P(\mathbf{x}, \mathbf{y})$ in polynomial time to the circuit's size (Choi et al., 2022).

Use the computation steps of $\max_{\mathbf{x}_h} \sum_{\mathbf{y}} P(\mathbf{x}_e, \mathbf{x}_h, \mathbf{y})$ as an example. For a PC node $v$ defined on $\mathbf{x}'_e \subseteq \mathbf{x}_e$ $\mathbf{x}'_h \subseteq \mathbf{x}_h$ and $\mathbf{y}' \subseteq \mathbf{y}$, let $P_v$ denote the distribution encoded by node $v$, and compute $UB(v) = \max_{\mathbf{x}'_h} \sum_{\mathbf{y}'} P_v(\mathbf{x}'_e, \mathbf{x}'_h, \mathbf{y}')$.

- Suppose $v$ is a leaf node over single variable $x \in \mathbf{x}_e$, then $UB(v) = P_v(x)$ since $|\mathbf{x}'_h| = |\mathbf{y}'| = 0$; Suppose $v$ is a leaf node over variable $x \in \mathbf{x}_h$, then $UB(v) = \max_x P_v(x)$; Suppose $v$ is a leaf node over variable $y \in \mathbf{y}$, then $UB(v) = \sum_y P_v(y) = 1$.

- Suppose $v$ is a product node. An example is shown in Figure 9. Without loss of generality, assume $v$ has 2 child nodes: $v_1$ and $v_2$ that encodes $P_{v_1}(\mathbf{x}_e^{(1)}, \mathbf{x}_h^{(1)}, \mathbf{y}^{(1)})$ and $P_{v_2}(\mathbf{x}_e^{(2)}, \mathbf{x}_h^{(2)}, \mathbf{y}^{(2)})$ respectively. The decomposability property indicates that all its child nodes share no common variable. So the maximum of the product can be computed as the product of the maximum. Specifically, we have

$$
\begin{aligned}
UB(v) &= \max_{\mathbf{x}'_h} \sum_{\mathbf{y}'} P_v(\mathbf{x}'_e, \mathbf{x}'_h, \mathbf{y}') \\
&= \max_{\mathbf{x}'_h} \sum_{\mathbf{y}'} P_{v_1}(\mathbf{x}_e^{(1)}, \mathbf{x}_h^{(1)}, \mathbf{y}^{(1)}) P_{v_2}(\mathbf{x}_e^{(2)}, \mathbf{x}_h^{(2)}, \mathbf{y}^{(2)}) \\
&= \max_{\mathbf{x}'_h} \sum_{\mathbf{y}^{(1)}} \sum_{\mathbf{y}^{(2)}} P_{v_1}(\mathbf{x}_e^{(1)}, \mathbf{x}_h^{(1)}, \mathbf{y}^{(1)}) P_{v_2}(\mathbf{x}_e^{(2)}, \mathbf{x}_h^{(2)}, \mathbf{y}^{(2)}) \\
&= \max_{\mathbf{x}_h^{(1)}} \sum_{\mathbf{y}^{(1)}} P_{v_1}(\mathbf{x}_e^{(1)}, \mathbf{x}_h^{(1)}, \mathbf{y}^{(1)}) \cdot \max_{\mathbf{x}_h^{(2)}} \sum_{\mathbf{y}^{(2)}} P_{v_2}(\mathbf{x}_e^{(2)}, \mathbf{x}_h^{(2)}, \mathbf{y}^{(2)}) \\
&= UB(v_1) \cdot UB(v_2)
\end{aligned}
$$

- Suppose $v$ is a sum node. Noted that the probabilistic circuit should be Q-deterministic w.r.t. all decision variables, including both query and evidence variables. An example is shown in Figure 9. Without loss of generality, assume $v$ has 2 child nodes $v_1$ and $v_2$ that encodes $P_{v_1}$ and $P_{v_2}$, and their weights are $w_1$ and $w_2$ respectively. The smoothness ensures that all its child nodes have the same scope of variables. The Q-determinism ensures that if all querying variables are assigned, only one of its child nodes will have a non-zero probability value.

$$
\begin{aligned}
UB(v) &= \max_{\mathbf{x}'_h} \sum_{\mathbf{y}'} P_v(\mathbf{x}'_e, \mathbf{x}'_h, \mathbf{y}') \\
&= \max_{\mathbf{x}'_h} \sum_{\mathbf{y}'} \left( w_1 P_{v_1}(\mathbf{x}'_e, \mathbf{x}'_h, \mathbf{y}') + w_2 P_{v_2}(\mathbf{x}'_e, \mathbf{x}'_h, \mathbf{y}') \right) \\
&= \max_{\mathbf{x}'_h} \left( \sum_{\mathbf{y}'} w_1 P_{v_1}(\mathbf{x}'_e, \mathbf{x}'_h, \mathbf{y}') + \sum_{\mathbf{y}'} w_2 P_{v_2}(\mathbf{x}'_e, \mathbf{x}'_h, \mathbf{y}') \right) \\
&= \max_{\mathbf{x}'_h} \left( \sum_{\mathbf{y}'} w_1 P_{v_1}(\mathbf{x}'_e, \mathbf{x}'_h, \mathbf{y}'), \sum_{\mathbf{y}'} w_2 P_{v_2}(\mathbf{x}'_e, \mathbf{x}'_h, \mathbf{y}') \right) \qquad \text{by Q-determinism} \\
&= \max \left( w_1 UB(v_1), w_2 UB(v_2) \right)
\end{aligned}
$$

Using the calculation defined above, we can recursively calculate $UB(r)$ for the root node $r$. Since $r$ encodes $P(\mathbf{x}_e, \mathbf{x}_h, \mathbf{y})$, the strict upper bound is calculated. Similar steps and proof can be generalized to the lower bound. Our proposed ULW follows the calculation steps shown above. The calculation requires only one traversal of the probabilistic circuit.

$\square$

# D  EXPERIMENT SETTING

## D.1  BASELINES

**Gibbs-SAA and BP-SAA** are approximate SMC solvers based on Sample Average Approximation. The marginal probability in the form of $\sum_y P(x, y)$ is approximated by samples from a sampler. More specifically, use the sampler to generate a set of samples $\{(x, y^{(i)})\}$ according to the distribution proportional to the $P(x, y)$. Then the estimation of the marginal probability is the sample average $\frac{1}{N} \sum_{y^{(i)}} P(x, y^{(i)})$ multiplied by the number of possible configurations of $y$, for binary

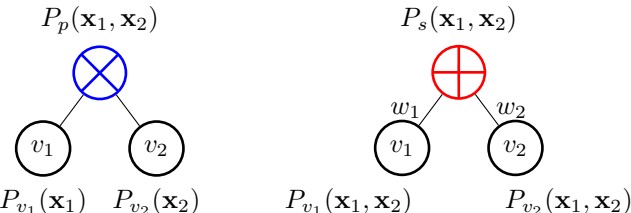

Figure 9: **(Left)** Example of a decomposable product node (colored blue). Denote the product node as $p$, and it has two children $v_1$ and $v_2$. Child nodes encode $P_{v_1}(\mathbf{x}_1)$ and $P_{v_2}(\mathbf{x}_2)$ respectively and the product node encodes $P_p(\mathbf{x}_1, \mathbf{x}_2) = P_{v_1}(\mathbf{x}_1)P_{v_2}(\mathbf{x}_2)$. Decomposability ensures $\mathbf{x}_1$ and $\mathbf{x}_2$ are disjoint. **(Right)** Example of a smooth and Q-deterministic (w.r.t. $\mathbf{x}_1, \mathbf{x}_2$) sum node (colored red). Denote the sum node as $s$, and it has two children $v_1$ and $v_2$ with weights $w_1$ and $w_2$. Child nodes encode $P_{v_1}(\mathbf{x}_1, \mathbf{x}_2)$ and $P_{v_2}(\mathbf{x}_1, \mathbf{x}_2)$ respectively and the sum node encodes $P_s(\mathbf{x}_1, \mathbf{x}_2) = w_1 P_{v_1}(\mathbf{x}_1, \mathbf{x}_2) + w_2 P_{v_2}(\mathbf{x}_1, \mathbf{x}_2)$. Smoothness ensures all nodes encode probabilities over the same set of variables, and Q-determinism means $P_{v_1}(\mathbf{x}_1, \mathbf{x}_2)$ and $P_{v_2}(\mathbf{x}_1, \mathbf{x}_2)$ can't be both non-zero under the same variable assignment.

variables of length $n$, there are $2^n$ possible configurations. We used Gibbs Sampler (Gibbs-SAA) and Belief Propagation (BP-SAA) implemented by (Fan & Yexiang, 2020) as the sampler. However, the sampling is only an efficient probability inference method, it still requires determining $x$ forehead, thus we use MiniSAT to enumerate solutions of $\phi(x)$. Given a time limit of 1 hour, we set the number of samples to 10000 and the number of Gibbs burn-in steps to 40. For each SMC problem in the benchmark dataset, we run Gibbs-SAA 5 times and the problem is considered "solved" as one of those runs produces a correct result. The percentage of solved SMCs is shown in Figure 3.

**XOR-SMC** is an approximate solver from (Li et al., 2024). We set the parameter $T$ (controlling the probability of a satisfying solution, a higher $T$ gives a better performance but longer run time) to 3, and incrementally increase the number of XOR constraints from 0 to either timeout or failed, by doing this we can find the most probable satisfying solution. Similar to SAA based approaches, we also run XOR-SMC 5 times.

**Lingeling-LibDAI and Lingeling-Toulbar2** are the integration of an SAT solver, Lingeling (Biere, 2017), with the winning probabilistic inference solver of UAI Approximate Inference Challenge. The procedure is first run Lingeling to produce one solution satisfying the Boolean formula in an SMC problem, then use the inference solver to calculate the marginal probability given those assignments. If the marginal probability exceeds the threshold, the solution is reported and exits. Otherwise, let the SAT solver produce another different solution and redo the procedure until all solutions have been enumerated. The repetitive file I/O and solvers' initialization time throughout the process have been pruned for a fair comparison. Lingeling-LibDAI uses the public inference solver implemented by LibDAI (Mooij, 2010) available on github[3]. Lingeling-Toulbar2 uses another inference solver Toulbar2 (Cooper et al., 2010) which uses a hybrid best-first branch-and-bound algorithm (HBFS) to solve marginal probability. We use the public implementation of Toulbar2[4] for PR task with their default parameters.

**Lingeling-D4, Lingeling-ADDMC, and Lingeling-SSTD** are integrations of the Lingeling SAT solver with the weighted model counting solver in the Model Counting Competition from 2020-2023. SAT-D4[5] uses d4 solver based on knowledge compilation. SAT-ADDMC uses the public implementation of ADDMC solver [6]. SAT-SSTD uses SharpSAT-TD[7] as the model counter.

---

[3]LibDAI: https://github.com/dbtsai/libDAI/
[4]Toulbar2: https://toulbar2.github.io/toulbar2/
[5]d4: https://github.com/crillab/d4
[6]ADDMC: https://github.com/vardigroup/ADDMC
[7]SharpSAT-TD: https://github.com/Laakeri/sharpsat-td

### D.2 Hyper-Parameter Settings

In all experiments, we use the public version of Lingeling implemented in PySAT[8] with their default parameter. The time limit for all approximate solvers (Gibbs-SAA, XOR-SMC) is set to 1 hour per SMC problem. The time limit for all exact solvers is 3 hours. All experiments are executed on two 64-core AMD Epyc 7662 Rome processors with 16 GB of memory.

### D.3 Dataset Specification

All SMC problems in this study are in the form of $\phi(\mathbf{x}_\phi, \mathbf{x}_f) \wedge \left( \sum_\mathbf{y} f(\mathbf{x}_f, \mathbf{y}) > Q \right)$ where $\phi(\mathbf{x}_\phi, \mathbf{x}_f)$ is a CNF Boolean formula, $f$ is a (unnormalized) probability distribution. $\mathbf{x}_\phi$ are variables appear only in $\phi$, $\mathbf{x}_f$ are random variables shared by $\phi$ and $f$, and $\mathbf{y}$ are variables to be marginalized.

**Boolean Formula** All $\phi(\mathbf{x}_\phi, \mathbf{x}_f)$ represent 3-coloring problems for graphs, which is to find an assignment of colors to the nodes of the graph such that no two adjacent nodes have the same color, and at most 3 colors are used to complete color the graph. Then each node in the graph corresponds to 3 random variables, says $x_1$ $x_2$ and $x_3$, that $x_1 = True$ iff. this node is colored with the first color. We consider only grid graphs of size $k$ by $k$, resulting in $k \times k \times 3$ variables.

Those Boolean formulas are generated by CNFgen[9] using the command

```
./cnfgen kcolor 3 grid k k -T shuffle
```

where the graph size $k$ is set to 5, 10, and 15. For each grid graph, we shuffle the variable names randomly and keep 3 of them.

**Probability Distribution** We use probabilistic graphical models from the UAI competition 2010-2022[10] including Markov random fields and Bayesian networks for the probabilistic constraints. Specifically, we pick the data for PR inference task, which includes 8 categories: Alchemy (2 models), CSP (3), DBN (6), Grids (8), ObjectDetection (79), Pedigree (3), Promedas (33), and Segmentation (6). The models with non-Boolean variables are removed, resulting in the remaining 50 models: *Alchemy* (1 model), *CSP* (3), *DBN* (6), *Grids* (2), *Promedas* (32), and *Segmentation* (6). All distributions are in the UAI file format. Since model counters d4, ADDMC, and SharpSAT-TD only accept weight CNF format in the model counting competition, we use bn2cnf[11] to convert data.

**Boolean Variables Classification** We pick random variables from $\phi$ and $f$ as shared variables uniformly at random. The number of shared variables between $\phi$ and $f$ (denoted as $\mathbf{x}_f$) is determined as the lesser of either half the number of random variables in $f$ or the total number of random variables in $\phi$, i.e., the count of variables in $\mathbf{x}_f$ will not surpass either half the total number of variables in $f$ or the entire count of variables in $\phi$.

### D.4 Supply Chain Design

For the experiment on real-world supply chain network data, we refer to a 4-layer supply chain network collected from real-world data (Zokaee et al., 2017). This network consists of 4 layers of nodes, representing suppliers, with each layer containing 9, 7, 9, and 19 nodes, respectively. Adjacent layers are fully connected, meaning each node can trade with any node in the adjacent layers (nearest upstream suppliers and downstream demanders). An example is shown in Figure 10. Each edge between two nodes represents a trade between them, and the selection of trades can be encoded as a binary vector $x \in \{0,1\}^M$, where $M$ is the number of edges. Here, $x[i] = 1$ indicates that the $i$-th edge (trade) is selected.

The original problem does not account for stochastic disasters, so we generated a random Bayesian Network (BN) over all edges to model such events. For example, $P(x_1 = \text{True}, x_2 = \text{False})$

---

[8]PySAT: https://pysathq.github.io/

[9]CNFgen: https://massimolauria.net/cnfgen/

[10]UAI2022: https://uaicompetition.github.io/uci-2022/

[11]bn2cnf: https://www.cril.univ-artois.fr/KC/bn2cnf.html

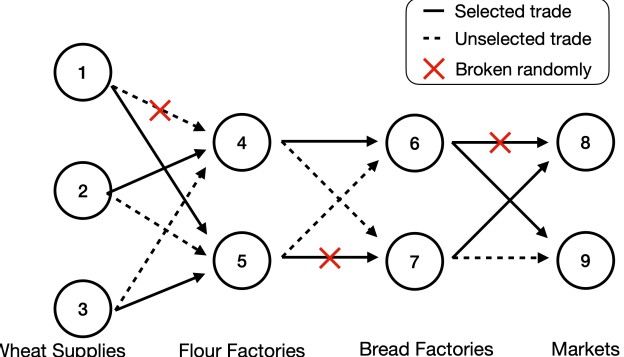

Figure 10: An example supply chain network. Edges with the red cross sign mean they are broken due to natural disasters.

represents the probability that trade 1 is successful while trade 2 fails. Each BN node can have at most 5 parents, and the number of BN edges is approximately half of the maximum possible number. The generated BN is included in the code repository.

Due to budget constraints, each node is assumed to receive raw materials from exactly 2 upstream suppliers and sells its product to exactly 2 downstream demanders. We want the probability that all trades are successfully conducted to be above a certain threshold, even in the face of random events such as natural disasters. We have the formulation:

$$\phi(\mathbf{x}, \mathbf{x}') \wedge \left( \sum_{\mathbf{x}'} P(\mathbf{x}, \mathbf{x}') > Q \right)$$

where $\phi(\mathbf{x}, \mathbf{x}')$ represents the plan of executing trades $\mathbf{x}$ while discarding $\mathbf{x}'$ to satisfy the budget constraints. The marginal probability $\sum_{\mathbf{x}'} P(\mathbf{x}, \mathbf{x}')$ is exactly the probability that all selected trades are carried out successfully. To find the optimal plan, we gradually increase the threshold $Q$ from 0 to 1 in increments of $1 \times 10^{-3}$, continuing until the threshold makes the SMC problem infeasible. The last feasible solution is referred to as the best plan.

We test all exact SMC solvers on 3 supply chain networks, including a small $[5, 5, 5, 5]$, a medium $[7, 7, 7, 7]$, and a large network $[9, 7, 9, 19]$. The vector $[9, 7, 9, 19]$ is the structure in the real world, representing a network with 9, 7, 9, and 19 suppliers in each layer, respectively. The other two networks are synthetic, but they have similar scales. The results are shown in Figure 6.

### D.5 Package Delivery Scheduling

For the case study of package delivery, our goal is to deliver packages to $N$ residential areas. We want this path to be a Hamiltonian Path that visits each vertex (residential area) exactly once without necessarily forming a cycle. The goal is to determine whether such a path exists in a given graph.

Using an order-based formulation with variables $x_{i,j}$, where $x_{i,j}$ denotes that the $i$-th position in the path is occupied by residential area $j$, i.e., residential area $j$ is the $i$-th visited place.

$$x_{i,j} = \begin{cases} \text{True} & \text{if area } j \text{ is visited in the } i\text{-th position in the path,} \\ \text{False} & \text{otherwise.} \end{cases}$$

where the total number of variables is $N^2$ (for $N$ cities).

To ensure that the variables $x_{i,j}$ correctly represent a valid Hamiltonian Path, several constraints must be enforced. These constraints formulate the Boolean satisfiability $\phi(x)$ in the SMC problem formulation.

- Each position is occupied by exactly one residential area. Or more formally, for every position $i$, exactly one residential area $j$ must occupy it.

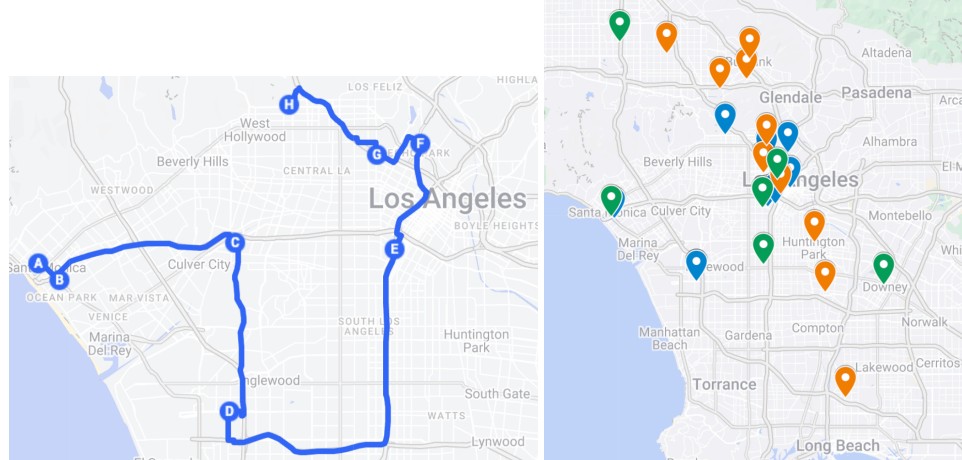

Figure 11: **(Left)** Example Hamiltonian delivery path covering major Amazon's lockers in Los Angeles. **(Right)** delivery locations included in experiments. Blue points are Amazon lockers, orange points are UPS stores, and green points are USPS stores.

- At least one residential area per position:

$$\bigvee_{j=1}^{n} x_{i,j} \quad \forall i \in \{1, 2, \ldots, n\}$$

- At most one residential area per position. For each position $i$ and for every pair of distinct areas $j$ and $k$:

$$\neg x_{i,j} \vee \neg x_{i,k} \quad \forall i, \forall j < k$$

- Each residential area appears exactly once in the path. Each residential area $j$ must be assigned to exactly one position $i$.
  - At least one position per residential area.

$$\bigvee_{i=1}^{n} x_{i,j} \quad \forall j \in \{1, 2, \ldots, n\}$$

  - At most one position per residential area: For each area $j$ and for every pair of distinct positions $i$ and $k$:

$$\neg x_{i,j} \vee \neg x_{k,j} \quad \forall j, \forall i < k$$

- Consecutive cities in the path are connected by an edge in the graph.
  - For each pair of consecutive positions $(i, i+1)$, the cities assigned must be connected by an edge. For all $i \in \{1, 2, \ldots, n-1\}$ and for all pairs of cities $(j, k)$ not connected by an edge in the graph:

$$\neg x_{i,j} \vee \neg x_{i+1,k} \quad \forall (j, k) \notin E$$

Additionally, we want the schedule to have a very high probability ($\geq Q$) of encountering light traffic.

$$P(\text{light traffic}|\text{path}) = \sum_{l} P(\text{light traffic}, l|\text{path}) \geq Q$$

where $l$ represents latent variables that affect the probability of traffic conditions, such as weather, road conditions, etc.

The graph structures used in our experiments are based on cropped regions from Google Maps (Figure 11). We consider three sets of delivery locations: 8 Amazon Lockers, 10 UPS Stores, and 6 USPS Stores. The three maps we examine are: Amazon Lockers only (Amazon), Amazon Lockers

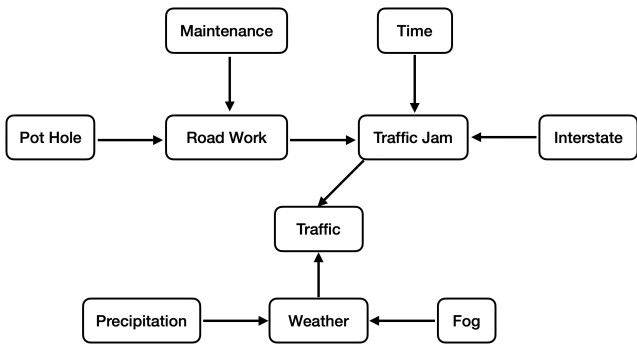

Figure 12: Bayesian network for a single road (Hoong et al., 2012; West, 2020).

plus UPS Stores (UPS), and UPS graph with the addition of 6 USPS Stores (USPS). These graphs consist of 8, 18, and 24 nodes, respectively.

The traffic condition probability is modeled by the Bayesian network (Figure 12) from Los Angeles traffic data (West, 2020). Instead of considering the "Time", we uses the order of traveling on a road to implicitly model the time.

To find the best route, we gradually decrease the threshold of the probability of encountering heavy traffic from 1 to 0 in increments of $10^{-2}$, continuing until the threshold makes the SMC problem unsatisfiable. The running time for finding the best plan is shown in Figure 6 (Right).

# E    ADDITIONAL RESULTS

## E.1    KNOWLEDGE COMPILATION TIME

The time for compiling graphical models to decomposable deterministic and smooth probabilistic circuits is shown in Figure 13. As shown in the subsequent additional plots, the knowledge compilation time most significantly affects the running time of probabilistic models from DBN and Segmentation.

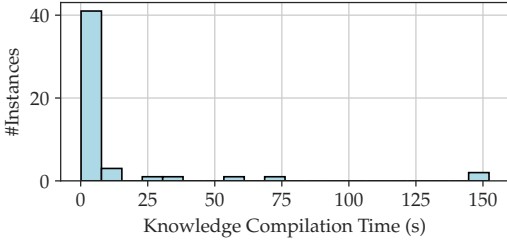

Figure 13: Histogram of the knowledge compilation time for all 50 probability distributions in the benchmark.

## E.2    COMPARISON OF DIFFERENT SAT SOLVERS

As an extension of Figure 4(Right), we also include MiniSAT (MINI-) and CaDiCal (CDC-) SAT solvers implemented in PySAT as the Boolean SAT oracle. The results are in Figure 14, KOCO-SMC shows the best performance among baselines.

## E.3    COMPARISON WITH EXACT SOLVERS

Figure 4 (Left) is one illustrating example shown in the main text. Additional results on other SMCs consisting of different Boolean formulas and probabilistic graphical models are shown below.

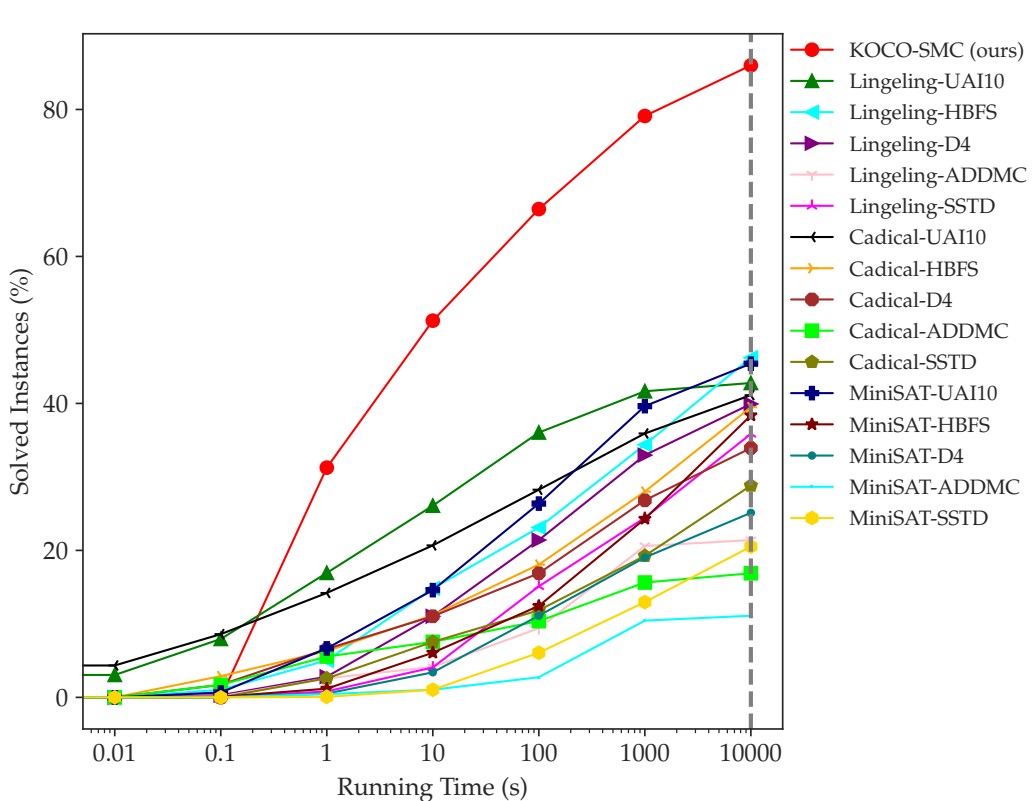

Figure 14: Running time of different exact solvers. Our KOCO-SMC solves more instances given the same amount of wall time.

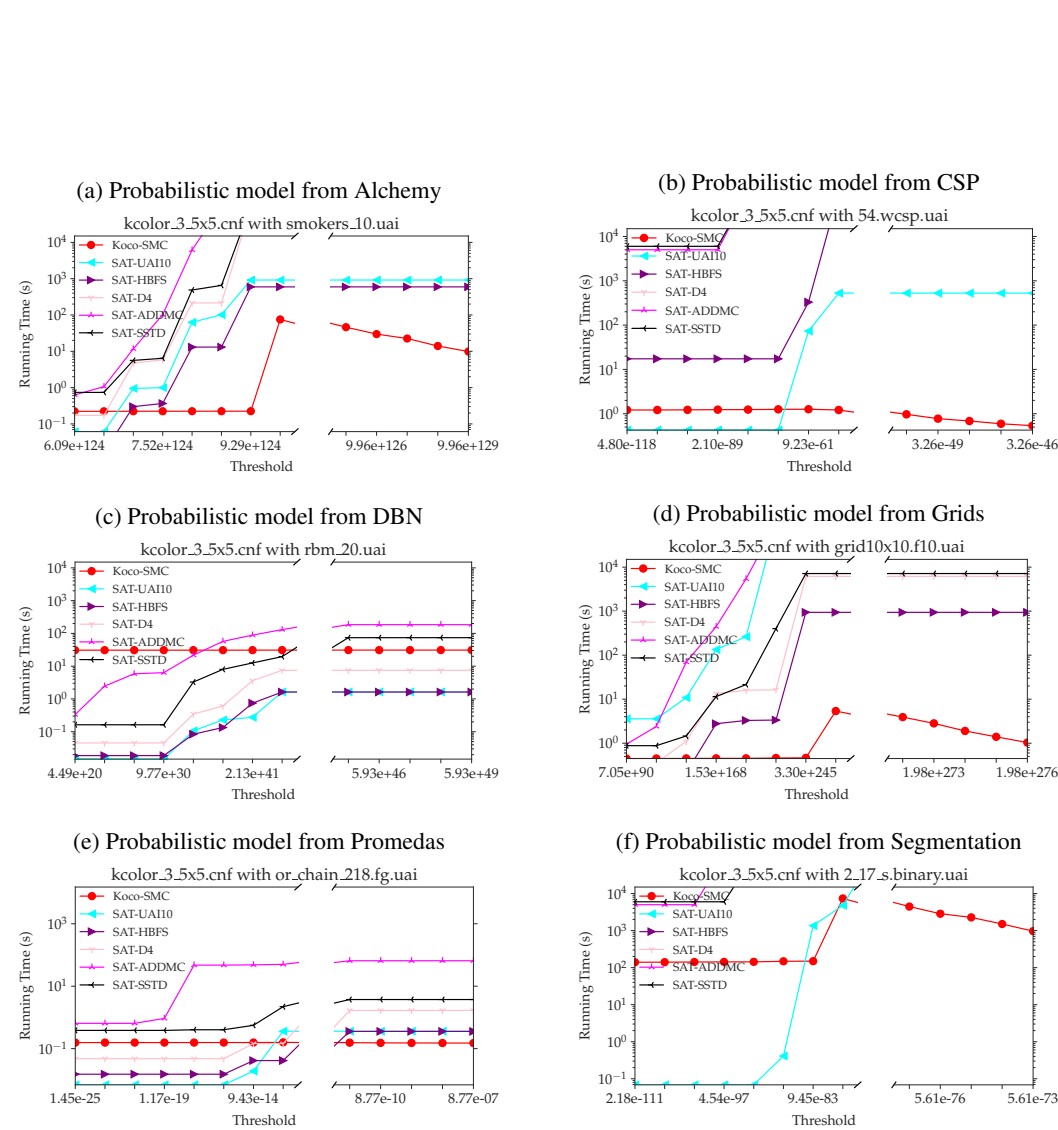

Figure 15: Results of SMC problems that consists of a fixed CNF file (*kcolor_3_5x5.cnf*) representing the 3 color problem on a $5 \times 5$ grid map and probabilistic graphical models from different categories.

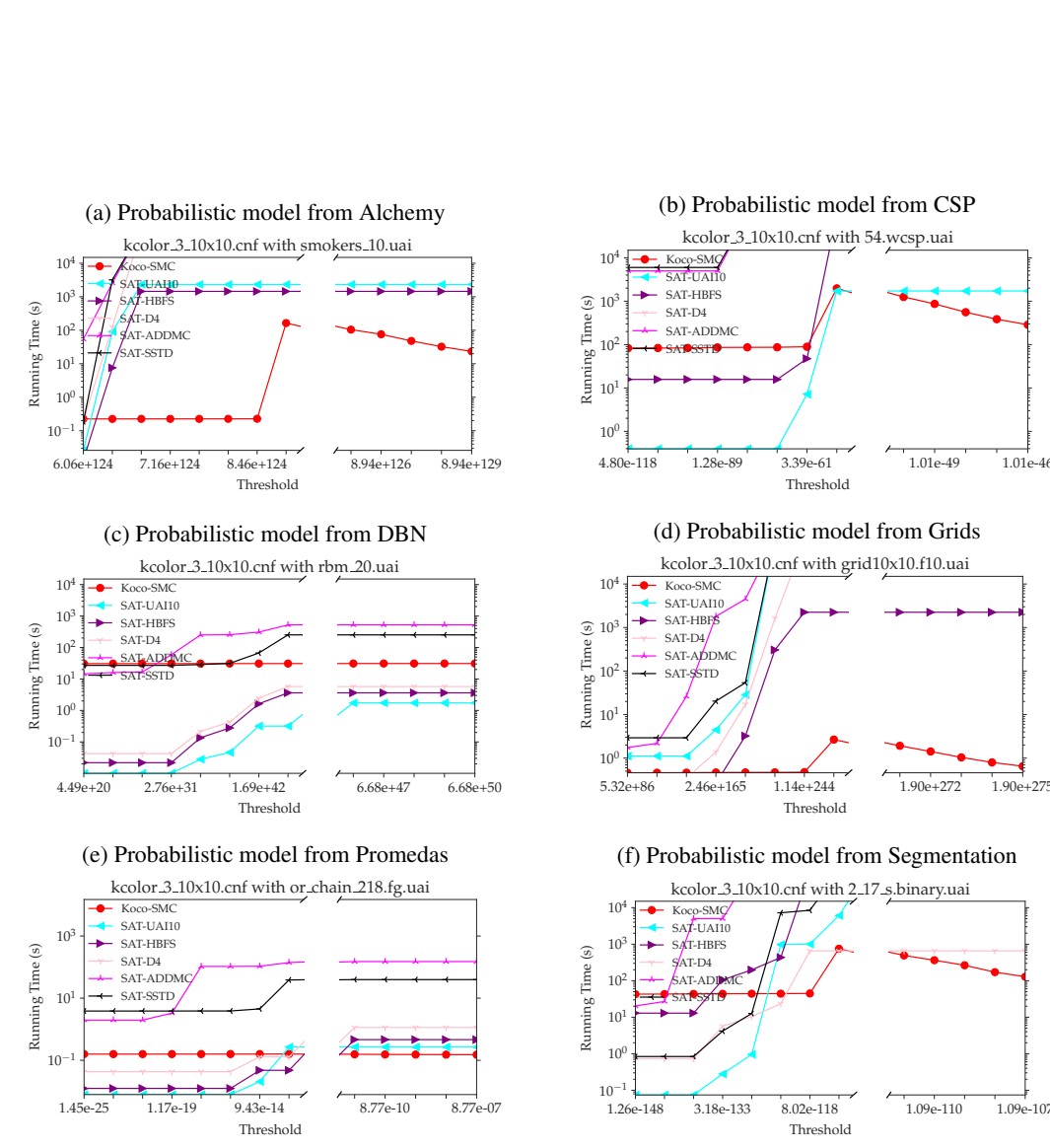

Figure 16: Results of SMC problems that consists of a fixed CNF file (*kcolor_3_10x10.cnf*) representing the 3 color problem on a $10 \times 10$ grid map and probabilistic graphical models from different categories.

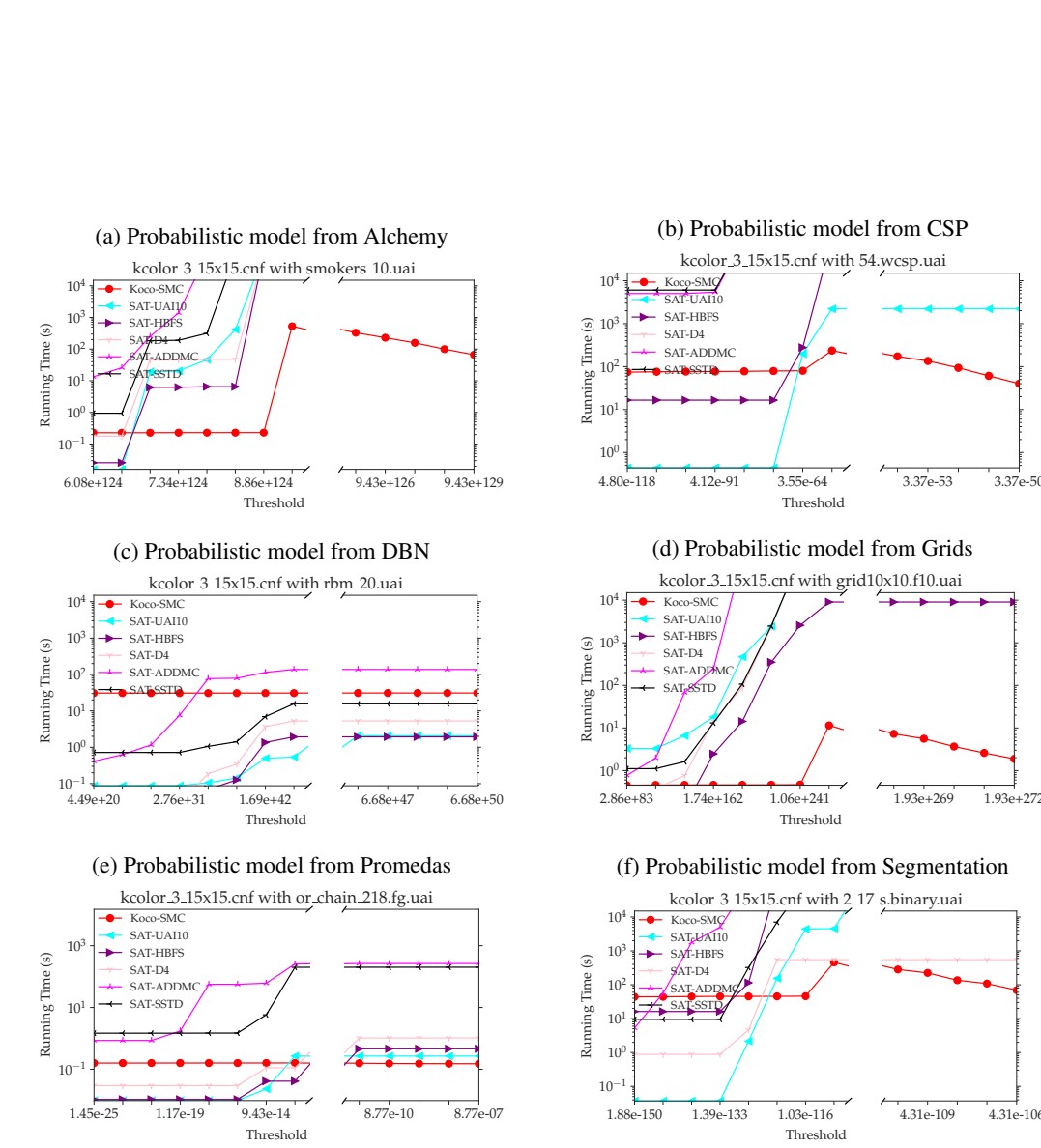

Figure 17: Results of SMC problems that consists of a fixed CNF file (*kcolor_3_15x15.cnf*) representing the 3 color problem on a $15 \times 15$ grid map and probabilistic graphical models from different categories.

