# OpenReview forum: "An Exact Solver for Satisfiability Modulo Counting with Probabilistic Circuits"
_ICLR.cc/2025/Conference — Submitted to ICLR 2025_

### Official Review · Reviewer_mJng · 2024-10-31

**Soundness:** 3
**Presentation:** 3
**Contribution:** 3
**Rating:** 6
**Confidence:** 3

**Summary:**

Satisfiability Modulo Counting (SMC) is an extension of SAT that incorporates constraints that involve model counting. Since probabilistic inference and model counting are two closely related concepts, SMC adaptively captures the satisfiability problem in scenarios involving uncertainty.

In this paper, the authors introduce an exact SMC solver (KOKO-SMC). The core idea is to track the upper and lower bounds of the probability inside probabilistic constraints during variable assignments. If these bounds violate the satisfaction condition the conflicts are recorded as learned Boolean clauses and appended to the Boolean part. This prevents the same conflict from occurring in future interactions.

Another feature of the approach is that knowledge compilation from discrete functions are integrated into probabilistic circuits. This speeds up the updates of bounds.

**Strengths:**

The paper presents an interesting approach for Satisfiability Modulo Counting. The authors also implement their approach in a new exact solver for SMC. The experimental results give some evidence that their solver performs better than other solvers in some relevant benchmarks.

**Weaknesses:**

The approach is strongly based on the conflict driven clause learning approach (CDCL) paradigm. So from a technical perspective, the main contribution seems to be the Upper-Lower Watch algorithm, which is an algorithm for monitoring the Marginal MAP of a probabilistic constraint using probabilistic circuits. I'm not sure how novel this approach is.

**Questions:**

Could you discuss related work on the Upper-Lower Watch approach to keep track of the values in the Marginal MAP? What is the novel step in this algorithm, how does it compare with existing approaches, etc?

---

> ### Author Response · Authors · 2024-11-24
>
> Thank you for your valuable review and feedback, which have provided us with an opportunity to clarify and strengthen our work.
>
> ------
> # 1. Main  contribution clarification
> Our main contribution is the proposal of an efficient exact solver for SMC problems, addressing a gap where existing solvers are either approximate or slow concatenation-based exact solvers. In the worst case, a concatenation-based exact solver would require enumerating almost all solutions before reaching an SAT/UNSAT conclusion.
>
> Our pipeline is built on the CDCL framework, as we recognized that probabilistic constraints can be seamlessly integrated into CDCL-based SAT solvers as additional "probabilistic clauses."
>
> ------
> # 2. Related work of the Upper-Lower Watch
> Our idea is to use the upper and lower bounds of probabilities to avoid unnecessary searches. Similar ideas can be found in branch-and-bound, which maintains upper and lower bounds of the objective function to prune suboptimal branches of the search tree, and in alpha-beta pruning, which uses bounds on the evaluation of game states to eliminate branches in the game tree that cannot affect the final decision. We have included this in the related work section (Paragraph 3, Appendix A).
>
> To make it a complete approach, we relate this bound monitoring to Marginal MAP inference and observe that probabilistic circuits with specific structures can enable efficient MMAP inference. These serve as the stepping stones of our approach.
>
> ------
> We hope this response addresses the reviewer’s concerns and provides clarity on the contributions. Thank you again for your insightful feedback.

---

### Official Review · Reviewer_Kf1C · 2024-11-03

**Soundness:** 2
**Presentation:** 2
**Contribution:** 1
**Rating:** 3
**Confidence:** 4

**Summary:**

This paper focuses on the design of a solver for  satisfiability modulo counting (SMC). SMC was introduced in 2014 by fredrikson and jha but have since received renewed interest over the past two years. Essentially, SMC allows counting predicates. The paper discusses a counting solver that combines CDCL methodology with lower and upper bounds via probabilistic circuits. The high-level description in the paper is not very different compared to   fredrikson and jha except for the usage of lower and upper bounds via probabilistic circuits.

**Strengths:**

On a high-level the paper is promising but my concerns are with empirical evaluation.

**Weaknesses:**

The empirical evaluation is limited to one specific setting which does not capture the class of SMC, in fact, the formulas can be encoded as stochastic SAT and therefore, one can rely on state of the art SSAT solvers such as [1].
Here is how we can encode the studied formulas into SSAT.

We can simply ask to solve Exists (X) Random (Y) \phi(X) \wedge f(X,Y).  -- note that this would provide a X for which the model count of f(X,Y) is maximized. The SSAT solvers are indeed aware of the exact count, and this can be retrieved from their logs. Otherwise, one can simply substitute the assignment of X and just run any of the state of the art model counters and compute counts.

There is one possibility that the aforementioned approach will not work but this requires authors to demonstrate empirically. I hope authors would do so during rebuttal phase and in such a case, I would be happy to revise my score to Accept.

[1] https://github.com/NTU-ALComLab/ssatABC

**Questions:**

Please see above.

---

> ### Author Response · Authors · 2024-11-24
> **The relevance and difference between he Stochastic SAT (SSAT) and SMC problems.**
>
> Thank you for raising the Stochastic SAT (SSAT) problems with a detailed explanation. We found the SSAT is highly relevant to our work. We would like to address your concerns as follows:
>
> # Summarized Message
>
> SSAT solvers can only handle SMC problems with one Boolean constraint and one probabilistic constraint. They do not generalize to those with multiple probabilistic constraints. Empirically, state-of-the-art SSAT solvers assume independence among random variables. In experiments, our probabilistic constraints are from real-world Markov random fields with highly correlated variables, where SSAT cannot be applied. Thus the comparison in efficiency would be limited.
>
> ------
> # Detailed explanation for SSAT and SMC problems
> -------
> ## 1. SSAT Captures a subset of SMC instances:
> The SMC problems with Boolean constraints and probabilistic constraints are outlined in Equation (1). In our experiments, we focused on a specific form of SMC problem:
> $\phi(X) \wedge \left(\sum_Y f(X, Y) > \text{threshold}\right)$, where X and Y are two sets of Boolean variables, $\phi(X)$ is a Boolean formula, and $\sum_Y f(X, Y)$ is a weighted model counting term. The goal is to determine whether there exists an assignment to X that satisfies the formula.
>
> SSAT solvers can address this specific case of SMC problems by leveraging their formulation of existential and randomized quantifiers. For the suggested SSAT problem: Exists(X), Random(Y), $ \phi(X) \wedge f(X, Y)$, where f(X, Y) is a weighted Boolean formula. Using state-of-the-art SSAT solvers, one could solve: $\arg \max_X \sum_Y f(X, Y)$, where X satisfies $\phi(X)$. Then we could equivalently solve the SMC problem mentioned before.
>
> SSAT solvers can efficiently handle this case of SMC problems. Yet, their applicability is limited with multiple probabilistic constraints. A comparison of the special case of SMC problems (with a single probabilistic constraint and independent variables) would provide only limited insight and would not generalize to the full class of SMC problems.
>
> -------
> ## 2. SSAT and SMC have a different specialization.
> - SSAT excels at handling problems with sequences of existential and randomized variables, such as those involving max-sum-max-... operations. This is beyond the scope of our SMC formulation.
>
> - SMC is suited to handling multiple probabilistic constraints simultaneously. For example: consider an SMC problem involving two probabilistic constraints $\phi(X) \wedge \left(\sum_Y f_1(X, Y) > q_1\right) \wedge \left(\sum_Z f_2(X, Z) > q_2\right)$. This type of scenario is common in real-world problems that SMC is designed to model. Modeling such problems as a max-sum problem would require optimizing multiple objectives over shared variables, which SSAT solvers cannot directly address.
>
>
> -------
> ## 3. The assumption on probability in SSAT is too rigid in SMC problem setting.
> A key limitation of SSAT solvers is their assumption of independence among all random variables $x_1, \ldots, x_n$, i.e., $P(x_1, \ldots, x_n) = P(x_1)P(x_2) \cdots P(x_n)$. This limits their applicability in scenarios where the joint distribution is complex. In contrast, KOCO-SMC uses challenging data with highly correlated variables modeled by Markov random fields.
> If we were to conduct experiments on these specific cases, we intuitively expect SSAT solvers to outperform KOCO-SMC due to the rich body of work and the many advances in the SSAT domain. However, these results would not reflect the broader applicability of KOCO-SMC to more general SMC problems.
>
> ------
> Thank you again for bringing up SSAT, as it represents an important and relevant line of work that we had not addressed. We have included a more detailed discussion of SSAT and its relationship to SMC in our revised manuscript (in Appendix A). We hope this addresses your concerns and look forward to any additional feedback you might have.

---

> > ### Comment · Reviewer_Kf1C · 2024-11-26
> >
> > Thanks for feedback. I did not mean to imply that SSAT captures SMC but we will need empirical evaluation to justify the need for SMC solvers. The instances that are reported are squarely within the realm of SSAT solvers.

---

### Official Review · Reviewer_rjKB · 2024-11-03

**Soundness:** 4
**Presentation:** 3
**Contribution:** 3
**Rating:** 8
**Confidence:** 4

**Summary:**

The paper studies an extended satisfiability problem involving both propositional and probabilistic inference and proposes an exact solver, KOCO-SMC, to this problem. The main idea is to integrate a Boolean SAT solver with a probabilistic circuit to detect the conflict in a probabilistic constraint early by monitoring the lower and upper bounds of its probability value. Experimental results over synthetic and real-world problem instances demonstrate the superior runtime performance of KOCO-SMC compared to the exact and approximate baselines.

**Strengths:**

1. The satisfiability modulo counting (SMC) problem is an emerging formulation that can capture symbolic constraints and probabilistic uncertainty simultaneously. The authors developed an efficient solver for the SMC problem which significantly outperformed the baselines.
2. The authors identified that the main weakness of the existing solvers is unable to detect the conflict in probabilistic constraints timely. They addressed it by using a probabilistic circuit to monitor the lower and upper bounds of the probability value for early conflict detection.
3. The paper presents the application of the SMC problem in real-world scenarios including supply chain design and package delivery problems, and shows the superior performance of applying their solver compared to the baselines.

**Weaknesses:**

1. The abused notation for $x_i$ in Section 3.1 confused me. For example, the sum over $x_3$ and $x_4$ at line 226 looks weird since they are already assigned to False at line 224.  I recommend using different variables to indicate the road selection and clearness.
2. The variable-watching scheme for probabilistic constraints misses some immediate conflicts. For example, we can not detect the conflict at line 233 timely if $x_1$ is assigned True but not watched.

**Questions:**

1. For the trivial combination of a SAT solver and a model counter, do you extract a conflict clause from a probabilistic constraint if the model counter finds it UNSAT?
2. In Figure 3, a problem instance is considered solved if one correct solution is found within five runs. Can you elaborate on how to define “solved” for an UNSAT instance?
3. What is the benchmark used in Figure 5? How did you calculate the running time of the left figure considering the unsolved instances? Did you consider the solved instances only or add a penalty for the unsolved instances?

---

> ### Author Response · Authors · 2024-11-24
>
> Thank you for your detailed comments and for giving us the opportunity to further clarify our paper.
>
> ## 1. The unclarity of current notation
> We will carefully revise the notations to ensure a clearer presentation in the next version. During the discussion period, we will keep it the same, because the changes in notation may cause misunderstandings for other reviewers.
>
> ------
> ## 2. Is there a conflict clause extracted from a probabilistic constraint?
>
> If the model counter identifies a solution that does not satisfy a probabilistic constraint, it adds a trivial clause negating the current assignment to the Boolean formula to avoid the same conflict. For example, if the Boolean SAT solver proposes the solution $(x_1 = \text{True}, x_2 = \text{False})$ and the model counter determines it is UNSAT, we add the clause $(\neg x_1 \vee x_2)$ to the Boolean formula.
>
> ------
> ## 3.  how to define “solved” in Figure 3 for UNSAT instances?
>
> For our KOCO-SMC, which is an exact solver, "solved" means that it determines the problem is UNSAT. For approximate solvers, "solved" means they successfully conclude UNSAT in at least one of five runs. In both cases, timeout instances are treated as UNKNOWN.
>
> ------
> ## 4. What is the benchmark in Figure 5?
>
> In Figure 5, we compare our KOCO-SMC with a variant that does not utilize the Upper-Lower Watch mechanism. In this variant, upper and lower bounds are not monitored for early conflict detection; instead, counting is performed only after all variables have been assigned. In the left figure, we illustrate the effect of varying the threshold by selecting a single SMC problem that remains solvable within 3 hours across different threshold values. Noted that “solvable” refers to reaching the correct answer (either SAT or UNSAT) in the given time.
>
> ------
> ## 5. Minor mistake
> We have revised the text around line 233.
>
> ------
> We hope the reviewer’s concerns are addressed and they will consider updating their score. We welcome further discussions.

---

> > ### Comment · Reviewer_rjKB · 2024-11-25
> >
> > Thanks for your reply. I don't have further questions.

---

### Official Review · Reviewer_F1o9 · 2024-11-03

**Soundness:** 2
**Presentation:** 2
**Contribution:** 2
**Rating:** 5
**Confidence:** 4

**Summary:**

The paper introduces KOCO-SMC, an exact solver for Satisfiability Modulo Counting (SMC) problems, integrating probabilistic circuits for efficient conflict detection. Unlike existing solvers, KOCO-SMC provides exact solutions efficiently by pre-compiling probabilistic inference. Key contributions include (1) KOCO-SMC as an efficient exact solver, (2) superior performance in experiments compared to other baselines, and (3) a case study showing its real-world applicability.

**Strengths:**

The paper’s strengths include KOCO-SMC’s innovative conflict detection for early pruning, which significantly speeds up SMC solving, especially in unsatisfiable cases. It effectively overcomes limitations of prior solvers, making it highly practical for real-world applications.

**Weaknesses:**

The paper's weaknesses include a limited problem scope, which restricts the broader applicability and generalizability of the results. Expanding the range of problem types could strengthen its impact. Additionally, the writing lacks readability.

**Questions:**

1. What is the complexity of KOCO-SMC?
   A complexity analysis would clarify its efficiency compared to existing SMC solvers.

2. How does KOCO-SMC differ from existing SAT-exact solvers?
   A detailed comparison could highlight KOCO-SMC’s unique conflict detection and pruning methods.

---

> ### Author Response · Authors · 2024-11-24
>
> Thank you for your valuable feedback. We collect the response to address your concerns as follows.
>
> # 1. For Limited Problem Scope
>
> The SMC framework is inherently general and capable of modeling a wide range of decision-making problems involving stochasticity. In the experimental section, we used instances from the UAI competition, which are from domains such as graph coloring, medical diagnosis, and family lineage analysis. We tried our best to cover as many application domains as possible.
>
> Furthermore, we demonstrated step-by-step how real-world cases can be formulated as SMC problems, such as route planning (Figure 2), supply chain design (Appendix C.4), and package delivery (Appendix C.5). These problems are then efficiently solved using KOCO-SMC. Our paper proposes an efficient exact solver for SMC problems, which can be applied to any SMC problem formulated as described in Equation (1).
>
> We would be delighted if you could suggest any additional relevant applications for the SMC framework.
>
>
> ------
> # 2. For Complexity Analysis
> The worst-case complexity of KOCO-SMC remains NP^PP, as finding an exact solution for SMC problems inherently falls into this complexity class. The complexity class NP^PP refers to problems solvable by an NP machine with access to an oracle for a PP (Probabilistic Polynomial-Time) problem. Our innovation lies in improving practical efficiency through pre-compilation and conflict detection rather than altering the theoretical complexity. We used experiments on synthetic data in Section 4.2 to demonstrate the efficiency.
>
> ------
> # 3. Comparison with Baseline Exact Solvers
>
> SAT-exact solvers require repeatedly finding a complete variable assignment of the Boolean part and verifying the model counting, which necessitates exploring every branch of variable assignments. In contrast, KOCO-SMC sequentially assigns variables one by one, detecting conflicts early and pruning unnecessary branches in the search space, significantly improving efficiency.
>
> We illustrate this with the example shown from line 221 to line 230.
> Baseline first computes an all-variable assignment, $x_1 = x_2 = b_1 = True, x_3=x_4=b_2=False$. Then the PC part computes the summation, which results in UNSAT.
> Our method immediately raises conflict given partial assignment $x_1 = \texttt{True}$, which saves time by avoiding assigning values to the rest variables. Because the PC part is UNSAT, i..e.,$P(x_1 = \texttt{True}, x_2, x_3, x_4)=0.1<0.5$, when $x_1 = \texttt{True}$.
>
> ------
> Thank you again for your feedback. We hope this clarifies our responses and addresses your concerns. Please let us know if you have other concerns.

---

### Official Review · Reviewer_KN3o · 2024-11-04

**Soundness:** 2
**Presentation:** 3
**Contribution:** 3
**Rating:** 3
**Confidence:** 4

**Summary:**

This paper proposes a method for solving satisfiability modulo counting (SMC) problems. SMC is a problem that consists of a combination of a Boolean SAT problem and a model counting problem. Current exact solvers combine a SAT solver with a model counting solver. However, this approach requires an excessive number of invocations of SAT and model counters and could be slow. The proposed KOCO-SMC solves SMC problems by using a probabilistic circuit and a SAT solver. By first compiling a probabilistic distribution into a circuit, it accelerates to answer probabilistic queries on solving an SMC problem. Moreover, using PCs enables the computation of upper and lower bounds of probabilities, which results in the efficient discovery of conflicts and contributes to efficient search. The paper evaluates the proposed approach extensively with real and synthetic datasets, showing that the proposed exact approach is more efficient than baseline methods.

**Strengths:**

1. The proposed method uses PCs effectively. PCs shine in situations where we need to answer different probabilistic queries for a distribution extensively. Speeding up SMC with a PC is a clever idea.
2. Extensive experimental results show the superiority of the proposed over multiple baseline methods on both synthetic and real data. Reading to significant improvements.
3. The presentation of the paper is clear and easy to understand the contributions.

**Weaknesses:**

Assumption 1 is not true. As shown by (Choi et al., 2022), smoothness, decomposability, and determinism do not enable tractable computation of MMAP query. Choi et al. (2022)  have shown that PCs satisfying Q-determinism (Definition 2) support tractable MMAP queries, but otherwise, it is difficult to answer MMAP in polytime. Since Q-determinism is the property specific for a given query set $Q$, answering MMAP for any query $Q$ is intractable. This was the motivation why Choi et al. (2022) proposed an iterative pruning method for MMAP.

If my understanding of the MMAP query is correct, Lemma 1 does not hold as is since it depends on Assumption 1. The paper should show why the proposed method can efficiently compute upper and lower bounds.

(Minor) Line 258: Unit propagation will not force $x_2 = False$ for this case.

**Questions:**

I'd be happy if the authors addressed the above concern about the MMAP query.

---

> ### Author Response · Authors · 2024-11-24
> **Q-determinism and its impact on our paper**
>
> We deeply appreciate your detailed review and for pointing out the issue in the Lemma regarding MMAP inference. Your observations are invaluable, and we address your concerns as follows:
>
> ------
> # 1. Clarification of the Requirement for Q-Deterministic Circuits
>
> You are correct. To guarantee exact upper and lower bound monitoring leveraging tractable MMAP inference, the probabilistic circuit must be Q-deterministic. We acknowledge that the previous version of our paper did not address this correctly. We have incorporated your comments and corrected the manuscript.
>
> The SMC problems we address can be represented as $\phi(x_1, \ldots, x_n) \wedge \left(\sum_y f(x_1, \ldots, x_n, y) > \text{threshold}\right)$, where $\phi(x_1, \ldots, x_n)$ is a Boolean formula, and $\sum_y f$ is a weighted model counting term. Denoting the set of decision variables $x_1, \ldots, x_n$ as the set $Q$, inferring the exact upper and lower bounds for $\sum_y f$ corresponds to solving an MMAP problem. This can be achieved by compiling the function $f$ into a Q-deterministic (smooth and decomposable) probabilistic circuit.
>
> During the solving process, even if some decision variables are assigned as the “evidence variable”, MMAP for the remaining unassigned variables remains tractable, as discussed in Section 2.2 of [Choi et al., 2022].
>
> However, in terms of experiments, we could not ensure that all our data could be compiled into Q-deterministic circuits due to limited control of the knowledge compiler. We have addressed this in the limitations section (Page 14, line 712). If Q-determinism could be guaranteed, the performance would likely improve further, due to the more accurate bound monitoring.
>
> ------
> # 2. Future extension: A Relaxed Upper-Lower watch without Q-Determinism
> Compiling distributions into Q-deterministic circuits can require extensive time or space (exceeding 20 GB in our experiments). To address this, we propose a relaxed version of ULW for broader applicability. The only modification lies in the updating rule for sum nodes. Specifically, when monitoring the upper bound at each sum node, instead of computing the maximum of its child nodes, we calculate the sum of their upper bounds.
>
> Intuitively, this guarantees that the value evaluated at each node is no less than the value after a full variable assignment, ensuring the validity of the upper and lower bounds. This approach relies only on decomposability and smoothness, eliminating the need for determinism. It may result in arbitrarily loose bounds so is not ideal for analysis. Due to time constraints, we will explore this relaxed version in greater detail with experimental results in the next iteration of our work.
>
> ------
>  Thank you again for your feedback. We hope this clarifies our responses and addresses your concerns. Please let us know if you have other concerns.

---

> > ### Comment · Reviewer_KN3o · 2024-11-24
> >
> > Thank you for the response. I like the idea of the paper and appreciate the authors' hard work in updating the manuscript in a short time. However, I think the paper would need another round to be accepted at a conference for the following reasons:
> >
> > 1.  As the authors mentioned, PCs satisfying Q-determinism could be much larger than those without Q-determinism. Since PC sizes affect both the pre-compilation time and the time required to answer each MMAP query, this change would have significant effects on the running time of the proposed method. However, the effect is not experimentally evaluated.
> > 2. If I understand correctly, the current experimental results are obtained with inappropriate estimations of upper and lower bounds. Therefore, it is possible that the proposed algorithm might fail to find an optimal solution.
> > 3. The new relaxed upper-lower watch method seems interesting and could be promising, but we need experimental validation.
> > 4. Since efficient MMAP is a key ingredient of the proposed method, modifying it would require a significant update of the paper. Therefore, I think the paper requires a thorough review by multiple reviewers,  but it is unrealistic to do it in this short discussion period.
> >
> > ## Question:
> > The authors say that we can use Q-determinism in the SMC problem we consider since they have a form of $\phi(x_1, \ldots, x_n) \wedge \left(\sum_y f(x_1, \ldots, x_n, y) > \text{threshold}\right)$. However, the motivating example of Eq. (2) is not in the form since the variables marginalized out in the counting terms ($x_1, \ldots, x_4$ in term (e), (f)) appear in logical constraints (term (b), (c)). Is it possible to construct a PC with Q-determinism for this example?

---

> > > ### Comment · Reviewer_KN3o · 2024-11-25
> > >
> > > I changed my score to 3. The idea of combining PCs with an SMT solver is promising, but I believe this paper needs another round to address the issues I pointed out in the above comment. I encourage the authors to submit a revised version to another high-level conference.

---

### Meta-Review · Area_Chair_f1sW · 2024-12-19

**Metareview:**

This paper provides a method for solving satisfiability modulo counting (SMC) problems. The particular contribution is to use a probabilistic circuit instead of a standard model counting solver.

While the approach is novel, this paper cannot be accepted in its current state. As the authors acknowledge, there was a mistake in Assumption 1, leading to consequences about the complexity of the method and requiring significant change in the experimental setup. The authors are encouraged to fix the experiments and resubmit to a new venue.

**Additional Comments On Reviewer Discussion:**

In the discussion with reviewer KN3o, it became clear that there was a mistake in the assumptions. Despite the clear effort of the authors to fix the paper, the reviewers agree that another reviewing round would be needed.

---

### Decision · Program_Chairs · 2025-01-22

Reject